Follow Up

# Optimization of systemic AAV9 gene therapy in Niemann–Pick disease, type C1 mice

Avani V Mylvara[1,2,*], Alana L Gibson[2,3,*], Tansy Gu[2,4,*], Cristin D Davidson[1,2], Arturo A Incao[2], Katerina Melnyk[1], Susan R Gembic[1], Dominick Pierre-Jacques[5], Stephanie M Cologna[5], Charles P Venditti[2], Forbes D Porter[1], William J Pavan[2]

Niemann–Pick disease, type C1 (NPC1), is a rare, fatal neurodegenerative disorder caused by pathological variants in *NPC1*, which encodes a lysosomal cholesterol transporter. FDA-approved treatments are limited and do not target the underlying genetic defect. Both systemic and central nervous system delivery of AAV9-h*NPC1* have shown significant disease amelioration in NPC1 murine models. To assess the impact of dose in null *Npc1^{m1N/m1N}* mice, we systemically administered three different doses of AAV9-h*NPC1* at 4 wk old. Then, to assess the impact of age, we administered the medium dose before phenotypic onset or at early or late stage of disease progression (4, 6, or 8 wk old, respectively). Higher vector doses and earlier treatment were associated with significantly increased lifespan, slower disease progression, and enhanced central nervous system transduction. In *Npc1^{I1061T/I1061T}* mice, a model recapitulating a common human hypomorphic variant, similar benefits ensued. Our findings help define dose ranges, treatment ages, and efficacy in hypomorphic models of NPC1 deficiency and suggest that higher doses of AAV9-h*NPC1* in presymptomatic disease states are likely to yield better outcomes in NPC1 individuals.

## Introduction

Niemann–Pick disease, type C (NPC), is a rare, fatal neurodegenerative disease with an incidence of ~1 in 100,000 live births [1]. This autosomal recessive lysosomal storage disorder is marked by unesterified cholesterol and sphingolipid accumulation in the lysosome, the latter of which is especially prominent in neural tissue. At least 95% of NPC1 individuals have disease-associated variants in the integral membrane-bound NPC1 protein located in the lysosome (NPC1 disease, OMIM #257220). The remaining individuals have pathological variants in the NPC2 protein (NPC2 disease, OMIM #607625), a soluble lysosomal protein that transfers unesterified cholesterol to NPC1 [2]. Clinical presentation of both forms of the disease is similar, where loss of function of NPC1 or NPC2 results in intracellular accumulation of unesterified cholesterol and glycosphingolipids [1, 3, 4, 5, 6, 7]. Disease severity and onset are highly variable, affecting infants, children, and adults; however, the classical presentation of NPC1 is most often observed in school-age children and typically includes progressive cerebellar ataxia, vertical supranuclear gaze palsy, gelastic cataplexy, motor deficits, and cognitive impairment, as well as visceral manifestations like hepatosplenomegaly [8, 9]. Miglustat, a glycosphingolipid synthesis inhibitor, is approved for treatment for NPC outside the United States. Until recently, there were no Food and Drug Administration (FDA)–approved therapies for NPC individuals. Then, in September 2024, the FDA approved arimoclomol (Miplyffa, in combination with miglustat) and levacetylleucine (Aqneursa) [10]. Though these and other investigational therapeutics provide some amelioration, all fail to address the root cause of the disorder—the absence of the functional NPC1 protein that leads to morbidity and mortality [11, 12, 13, 14, 15]. Gene therapy can provide replacement of the dysfunctional NPC1 protein to treat the disease [16, 17].

Many studies have demonstrated that adeno-associated viral (AAV) gene therapy can successfully treat monogenic and rare diseases in preclinical models [18, 19, 20, 21, 22, 23]. AAVs are already approved for treatment of individuals with disorders such as spinal muscular atrophy (SMA1; onasemnogene abeparvovec-xioi [24, 25, 26]), RPE65 mutation–associated retinal dystrophy (voretigene neparvovec-rzyl [27, 28]), and aromatic L-amino acid decarboxylase (AADC; eladocagene exuparvovec-tneq [29]). AAV9 is well documented to cross the blood–brain barrier (BBB) and transduce cells of the central nervous system (CNS) [30, 31, 32, 33, 34]. Given the devastating neurological impact of NPC1, gene

[1]Department of Human Health and Services, Eunice Kennedy Shriver National Institute of Child Health and Human Development, National Institutes of Health, Bethesda, MD, USA   [2]Department of Human Health and Services, National Human Genome Research Institute, National Institutes of Health, Bethesda, MD, USA   [3]Howard Hughes Medical Institute, Department of Cellular and Molecular Medicine, Section of Neurobiology, Division of Biological Sciences, University of California San Diego, San Diego, CA, USA   [4]University of North Carolina, Chapel Hill, NC, USA   [5]University of Illinois Chicago, Chicago, IL, USA

Correspondence: cristin.davidson@nih.gov; fdporter@mail.nih.gov
*Avani V Mylvara, Alana L Gibson, and Tansy Gu contributed equally to this work

therapy targeting the CNS is imperative. Of note, AAV9 also transduces multiple other organ systems, including the liver and peripheral nerves, that are implicated in NPC1 disease (8, 9, 35, 36, 37, 38, 39, 40, 41).

We and others have previously demonstrated that AAV9 vectors can effectively improve survival and delay disease progression in the null $Npc1^{m1N/m1N}$ ($Npc1^{m1N}$, single allele notation for homozygosity) murine model of NPC1 deficiency (18, 42, 43, 44, 45, 46). $Npc1^{m1N}$ mice have a premature stop codon in the $Npc1$ gene, leading to the production of truncated, nonfunctional NPC1 protein and resulting in a severe disease phenotype (18, 42, 43, 44, 45, 46, 47, 48, 49, 50, 51). Both direct CNS administration (intracerebroventricular or intracisterna magna) (43, 45) and systemic administration (retro-orbital or intracardiac) (18, 42, 44) have successfully ameliorated disease in these mice. Greater success has been noted when using dual routes of CNS administration or administering higher doses of vector to the CNS (45, 46). We and others have demonstrated that ubiquitous promoters provide greater disease correction in $Npc1^{m1N}$ mice as compared to neuron-specific promoters (18, 46). Further optimization studies have highlighted the therapeutic potential of novel capsids to improve CNS transduction (44).

Many preclinical gene therapy studies in $Npc1^{m1N}$ mice have targeted neonates (42, 43, 45, 46). Previously, we administered AAV9 vectors at 4 wk old, before onset of phenotypic signs (18, 44). The question remains whether late(r) intervention can still be effective after diagnoses in individuals after disease onset. Previous clinical studies for AADC deficiency, a rare pediatric genetic disorder, suggest that AAV gene therapy is universally beneficial, but treatment at a younger age was associated with greater benefits (29). Intervention before clinical onset of neurological symptoms in NPC is currently challenging because NPC1 is not routinely screened for in newborns and the average diagnostic delay remains ~4.1 yr (52, 53, 54, 55). However, early intervention before neurological onset might be possible in familial cases and after diagnosis when there is infantile presentation with fetal ascites and liver disease (1, 56, 57, 58).

More than 600 pathogenic or likely pathogenic NPC1 variants have been described, most of which are missense mutations (59, 60, 61). One of the most prevalent variants is a missense mutation resulting in an amino acid substitution (p.I1061T). The NPC1 p.I1061T protein misfolds and undergoes endoplasmic reticulum–associated degradation (ERAD) (62, 63, 64). A knock-in, hypomorphic $Npc1^{I1061T}$ allele was generated to recapitulate the human disorder and results in a slightly protracted disease course compared with the more severe $Npc1^{m1N}$ mouse model (65). To assess whether residual NPC1 protein with compromised stability affected gene therapy treatment, we evaluated efficacy of gene therapy in $Npc1^{I1061T/I1061T}$ mice ($Npc1^{I1061T}$, single allele notation for homozygosity). We investigated and confirmed that residual NPC1 protein with compromised stability did not interfere with the efficacy of gene therapy.

In this study, we build on our previous work (18, 44), using systemic administration of an AAV9-elongation factor 1α (shortened)-h$NPC1$ (AAV9-EF1a(s)-h$NPC1$) vector to treat NPC1 mice. We examine the therapeutic efficacy of this vector across different doses, at time points later in disease progression, and in $Npc1^{I1061T}$

mice. Our results provide foundational preclinical data for the advancement of AAV9-EF1a(s)-h$NPC1$ as a disease-modulating therapy for individuals with NPC1 deficiency.

# Results

## Dose selection and administration paradigm

Dose selection was based on our previously published work (18, 44), with vector genomes (vg) administered per mouse as the primary parameter. For consistency with other research and clinical trials, doses are also provided in terms of vg/kilogram (vg/kg). The medium dose administered at 4 wk (1.2 × $10^{12}$ vg or 1.28 × $10^{14}$ vg/kg) served as the baseline comparator across all studies. Lower and higher doses were chosen to capture a comprehensive dose–response range, considering both experimental and practical constraints, which include the vector concentration coupled with our Animal Care and Use Committee's approved volumes for retro-orbital (RO) delivery. The medium dose was selected for the age at treatment study to ensure sufficient vector was available for full study enrollment.

## $Npc1^{m1N}$ mice treated with higher doses of AAV9-EF1a(s)-h$NPC1$ show increased survival and delayed disease progression

All mice received a single RO injection of AAV9-EF1a(s)-h$NPC1$ (referred to as AAV9), and each figure panel following includes data from 4-28 mice per group. Regarding statistical analyses, no significant differences between sex were found for any readout measure in the dose, age at treatment, or hypomorphic Npc1 model substudies. Thus, data from males and females are combined in all statistical tests. The only exception to sex differences is weight curves, which were not analyzed. In this section, mice were euthanized either at 10 wk for an age-matched cohort or at humane endpoint for survival.

To compare the efficacy of AAV9 at different doses, mice were injected at 4 wk old with either low (1 × $10^{11}$ vg or 7.87 × $10^{12}$ vg/kg), medium (1.2 × $10^{12}$ vg or 1.28 × $10^{14}$ vg/kg), or high (4.3 × $10^{12}$ vg or 3.06 × $10^{14}$ vg/kg) dose. A log-rank, Mantel–Cox test, with Bonferroni's correction for multiple comparisons, was used to assess survival (Fig 1A, Table S1A). High- and medium-dose mice had a longer median survival (34.6 and 21.5 wk, respectively) compared with low-dose (11.4 wk) and saline-injected (10.6 wk) mice (for all, $P < 0.0001$). Notably, even the low-dose group had improved survival compared with the saline-injected group (11.4 versus 10.6 wk, $P = 0.005$).

To determine the effect of AAV9 dose on neurological disease progression, a composite phenotype score was assessed at 3-wk intervals from 6 to 18 wk, where increasing scores indicate disease progression (Fig 1B) (66). The five neurological phenotypic parameters evaluated for the composite score include hindlimb clasp, motor function, kyphosis, balance, and grooming. High-dose mice showed the greatest delay in disease progression, followed by medium-dose mice, whereas low-dose mice mirrored the

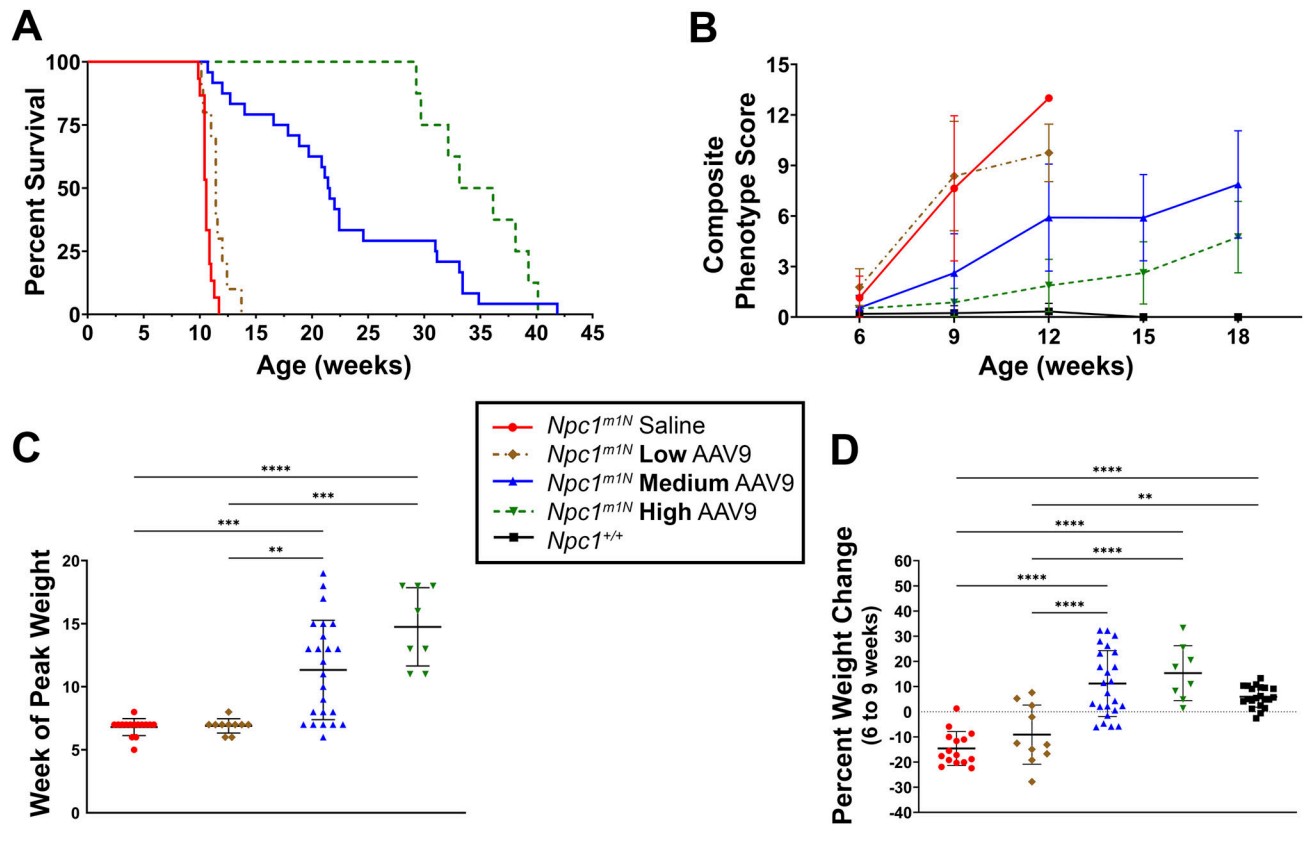

**Figure 1.** *Npc1^{m1N}* mice treated with AAV9-EF1a(s)-h*NPC1* vector show increased survival and delayed disease phenotype progression.
**(A)** Kaplan–Meier survival curve of mice treated with low-, medium-, and high-dose AAV9 and saline-injected mice (results presented in Table S1A) (saline, n = 15; low, n = 10; medium, n = 24; high, n = 8). **(B)** Composite phenotype scores for each dosage group with measurements taken every 3 wk, starting at 6 wk (results presented in Table S2A) (saline, n = 14; low, n = 10; medium, n = 13; high, n = 8; *Npc1^{+/+}*, n = 21). **(C)** Week at which mice reached peak weight (Kruskal–Wallis with Dunn's multiple comparisons test). **(D)** Percent weight change between 6 and 9 wk old (one-way ANOVA with Tukey's multiple comparisons test). For (C, D): saline, n = 15; low, n = 10; medium, n = 24; high, n = 8; for (D): *Npc1^{+/+}*, n = 21. For all: \*P < 0.05, \*\*P < 0.01, \*\*\*P < 0.001, \*\*\*\*P < 0.0001. **(B, C, D)** Data are presented as the mean ± SD for (B, C, D).

saline-injected mice (Fig 1B). Two-way ANOVA with mixed-effects analysis and Tukey's multiple comparisons test was used to assess differences between groups (Table S2A). Between 6 and 9 wk, the saline and low-dose groups had similar phenotype scores (P = 0.95), whereas the medium-dose and high-dose groups had significantly lower phenotype scores compared with the saline group (medium dose: P = 0.020; high dose: P < 0.0001) and the low-dose group (medium dose: P = 0.007; high dose: P < 0.0001). Notably, high-dose mice were not significantly different from *Npc1^{+/+}* mice (P = 0.91). From 9 to 12 wk, medium- and high-dose mice continued to show significantly lower phenotype scores than saline-injected and low-dose mice (for all, P < 0.0001). High-dose mice maintained lower phenotype scores closer to *Npc1^{+/+}* mice (P = 0.51). From 15 to 18 wk, high-dose mice had significantly lower phenotype scores than medium-dose mice (P = 0.008) and did not differ significantly from *Npc1^{+/+}* mice (P = 0.11).

*Npc1^{m1N}* mice exhibit marked weight loss starting at about 6 wk old. To evaluate disease onset, we assessed the week that mice reached peak weight (Fig 1C), and to evaluate disease progression, we assessed change in weight from 6 to 9 wk (Fig 1D). Low-dose mice reached peak weight at a similar time to saline-injected mice

(6.9 ± 0.6 wk and 6.8 ± 0.7 wk, respectively). In contrast, medium-dose mice reached peak weight significantly later (11.3 ± 3.9 wk) than saline-injected (P = 0.0007) or low-dose mice (P = 0.007). Similarly, high-dose mice reached peak weight significantly later (14.8 ± 3.1 wk) than saline-injected (P < 0.0001) or low-dose mice (P = 0.0003). Statistical significance for week of peak weight was determined using a Kruskal–Wallis test with Dunn's multiple comparisons test. Longitudinal weight data further demonstrate that mice maintain weight and survive longer as the dose of AAV9 increases (Fig S1A and B; Table S3A). When assessing percent weight change from 6 to 9 wk (Fig 1D), both saline-injected and low-dose mice showed similar percent weight loss (−14.6% ± 6.8% and −9.0% ± 11.7%, respectively; P = 0.64) (Fig 1D). In contrast, medium-dose (11.2% ± 13.1%) and high-dose (15.3% ± 10.9%) mice showed significant weight gain compared with both saline-injected and low-dose groups (for all, P < 0.0001). Medium-dose and high-dose mice gained weight similar to *Npc1^{+/+}* mice (5.6% ± 4.1%) during this period (medium dose: P = 0.38; high dose: P = 0.15). Statistical significance for weight change from 6 to 9 wk was assessed using one-way ANOVA with Tukey's multiple comparisons test.

## Higher doses of AAV9 enhance viral transduction in brain and liver of $Npc1^{m1N}$ mice

To evaluate the efficacy of vector transduction across tissues, droplet digital PCR (ddPCR) was performed on cerebellar (Fig 2Ai, iii), cerebral (Fig 2Bi, iii), and liver tissues (Fig 2Ci, iii) to assess h$NPC1$ copy number at 10 wk of age or at humane endpoint/ survival. In parallel, Western blots were performed to measure NPC1 protein levels present in cerebellum (Fig 2Aii), cerebrum (Fig 2Bii), and liver (Fig 2Cii), as NPC1 protein presence is exclusively attributed to vector transduction in $Npc1^{m1N}$ mice.

At 10 wk old, high-dose mice exhibited higher h$NPC1$ copy numbers in the cerebellum (Fig 2Ai), cerebrum (Fig 2Bi), and liver (Fig 2Ci) compared with all other groups. Similar trends were observed in the spleen, kidney, lung, muscle, and brain stem (Fig S2).

In the cerebellum (Fig 2Aii) and cerebrum (Fig 2Bii), NPC1 protein levels were low, but detectable in high-dose mice and $Npc1^{+/+}$ mice, with higher levels observed in $Npc1^{+/+}$. In the liver (Fig 2Cii), NPC1 protein was detected only in $Npc1^{+/+}$, high-dose, and medium-dose mice, with high-dose mice showing higher average NPC1 protein expression than $Npc1^{+/+}$ mice. Representative Western blots are shown in Fig S3 (cerebellum: Fig S3A and B; cerebrum: Fig S3C and D; liver: Fig S3E and F).

In a linear regression analysis, lifespan was significantly associated with h$NPC1$ copy number in the cerebellum and cerebrum within the medium-dose group (Fig 2Aiii and Biii). However, two mice with exceptionally high copy numbers in these regions drove significance. In the liver, copy number did not predict lifespan in any individual treatment group (Fig 2Cii, linear regression analysis).

## Higher doses of AAV9 reduce cerebellar and hepatic pathology in 10-wk-old $Npc1^{m1N}$ mice

Cerebellar ataxia is a major clinical feature in NPC individuals and is recapitulated in NPC1 mice with cerebellar pathology and Purkinje neuron degeneration (8, 51, 67). To assess cholesterol storage and pathological alterations, we performed immunofluorescence staining of the cerebellum in 10-wk-old mice. Parallel Western blot analysis was used for quantification.

Unesterified cholesterol storage, as visualized by filipin labeling (68), is increased in NPC1 disease (69). However, high-dose gene therapy appears to moderately reduce cholesterol storage compared with other treated or saline-injected mice (Fig 3A). Reactive astrocytosis (GFAP-positive staining) is a characteristic finding in both human and mice with NPC1 disease (70, 71, 72). Treatment with AAV9 shows a mild dose-dependent decrease in astrocytosis (Fig 3A). GFAP protein levels do not appear to differ between saline-injected, and low-, medium-, or high-dose mice (Fig 3C; representative blot in Fig S4A and B).

Progressive loss of Purkinje neurons, particularly from anterior to posterior cerebellar lobules, is a hallmark of NPC1 disease (51, 67) and one with which our data align. Purkinje neuron survival (calbindin D labeling) appears to improve with increasing doses of AAV9 and correlates with h$NPC1$ copy number. Nevertheless, the presence of Purkinje neuron remains markedly reduced compared with healthy $Npc1^{+/+}$ mice (Fig 3A). A similar dose-dependent

increase is observed with calbindin D protein levels, though levels remain below those of $Npc1^{+/+}$ mice (Fig 3D; representative blot in Fig S4D and E).

Microglial activation, a well-documented feature of NPC1 pathology (72, 73), appears mildly reduced in high-dose mice, particularly in posterior cerebellar lobules as shown by IBA1 staining (Fig 3B). This reduction is accompanied by a decrease in CD68 protein levels in high-dose mice, a marker of reactive microglia (74) (Fig 3E; representative blot in Fig S4B and C).

Hepatomegaly is a common feature of NPC1 disease, with infantile presentation often associated with liver disease (53, 57, 58). To evaluate liver involvement, we performed immunohistochemical (Fig 4A) and immunofluorescence (Fig 4C) staining of macrophages using CD68 labeling (75) in 10-wk-old mice. High- and medium-dose AAV9-treated mice showed a reduced percentage of CD68+ area compared with low-dose and saline-injected mice (Fig 4B; representative images in Fig 4A). In addition, immunofluorescence staining demonstrated a decrease in cholesterol storage (filipin labeling) and macrophage presence (CD68) as the dose of AAV9 increased (Fig 4C).

## AAV9 modulates sphingolipid accumulation in the brains of 10-wk-old $Npc1^{m1N}$ mice

In addition to cholesterol accumulation, multiple lipid classes exhibit altered levels in the NPC1-deficient brain because of impaired NPC1 protein function in mice (76, 77, 78). Among these, gangliosides such as GM2 accumulate in the brains of $Npc1^{m1N}$ mice (79). Mass spectrometry imaging was used to assess the impact of AAV9 dose on sphingolipid distribution in 10-wk-old $Npc1^{m1N}$ mice and $Npc1^{+/+}$ mice (Fig S5). Although $Npc1^{+/+}$ mice demonstrated little to no ganglioside accumulation, $Npc1^{m1N}$ mice displayed high accumulation in the frontal cortex and cerebellar lobule X. Increasing AAV9 doses led to a qualitative reduction in ganglioside accumulation in both regions (Fig S5Ai). However, quantitative analysis of GM2 (d36:1) levels did not reveal statistically significant differences between groups (Fig S5B).

Beyond gangliosides, sphingolipid distribution is broadly disrupted in $Npc1^{m1N}$ mice, with altered distribution of hexosylceramides and dihydroceramides in the cerebellum (80). Hexosylceramide (HexCer 46:4;O3) levels appeared lower in the cerebellum of $Npc1^{m1N}$ mice compared with $Npc1^{+/+}$ mice. AAV9 treatment was associated with a qualitative increase in HexCer abundance in a dose-dependent manner, particularly in the rostral cerebellar lobules (lobules I–V) (Fig S5Aii and Bii). Conversely, dihydroceramide (Cer 32:2;O3) levels appeared elevated in the cerebellum of $Npc1^{m1N}$ mice compared with $Npc1^{+/+}$ mice. Although AAV9 had a less pronounced effect on dihydroceramide accumulation, a visual trend toward reduction was observed at medium and high doses (Fig S5Aiii).

## AAV9 treatment at 4 wk improves survival and delays disease progression in $Npc1^{m1N}$ mice compared with treatment at 6 or 8 wk of age

$Npc1^{m1N}$ mice were treated with $1.28 \times 10^{14}$ vg/kg of AAV9 at 4 wk (before onset of neurological symptoms, hereafter referred to as

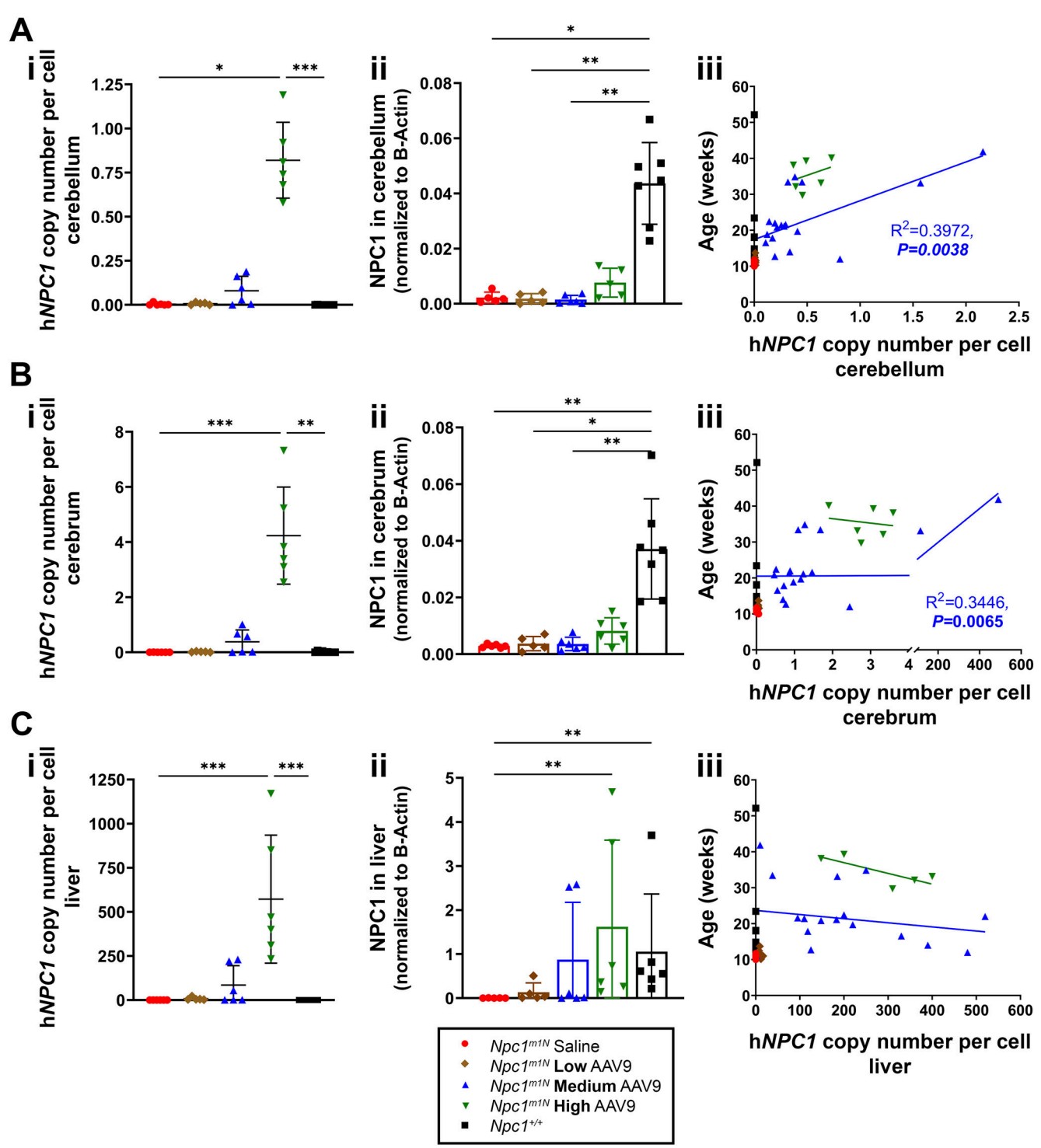

**Figure 2. Higher doses of AAV9 enhance viral transduction in the brain and liver of *Npc1^{m1N}* mice.**
**(A, B, C)** Analysis of cerebellum (A), cerebrum (B), and liver (C). **(Ai, Bi, Ci)** h*NPC1* copy number in cerebellum (Ai), cerebrum (Bi), and liver (Ci) from 10-wk-old mice (Kruskal–Wallis test with Dunn's multiple comparisons test). For (Ai, Bi, Ci): saline, n = 6; low, n = 5; medium, n = 6; high, n = 6; *Npc1^{+/+}*, n = 9. **(Aii, Bii, Cii)** NPC1 protein levels were assessed via Western blot in 10-wk-old mice to confirm the amount of NPC1 protein in the cerebellum (Aii), cerebrum (Bii), and liver (Cii) (Kruskal–Wallis test with Dunn's multiple comparisons test). For (Aii): saline, n = 6; low, n = 5; medium, n = 6; high, n = 6; *Npc1^{+/+}*, n = 8. For (Bii, Cii): saline, n = 6; low, n = 5; medium, n = 6; high, n = 6; *Npc1^{+/+}*, n = 7. **(Aiii, Biii, Ciii)** Linear regression of lifespan as a function of h*NPC1* copy number in the cerebellum (Aii), cerebrum (Bii), and liver (Cii). For (Aiii, Biii): saline, n = 14; low, n = 9; medium, n = 20; high, n = 6; *Npc1^{+/+}*, n = 17. For Ciii: saline, n = 14; low, n = 9; medium, n = 18; high, n = 5; *Npc1^{+/+}*, n = 17. For all: *$P < 0.05$, **$P < 0.01$, ***$P < 0.001$, ****$P < 0.0001$. **(Ai, Bi, Ci, Aii, Bii, Cii)** Data are presented as the mean ± SD for (Ai, Bi, Ci, Aii, Bii, Cii).

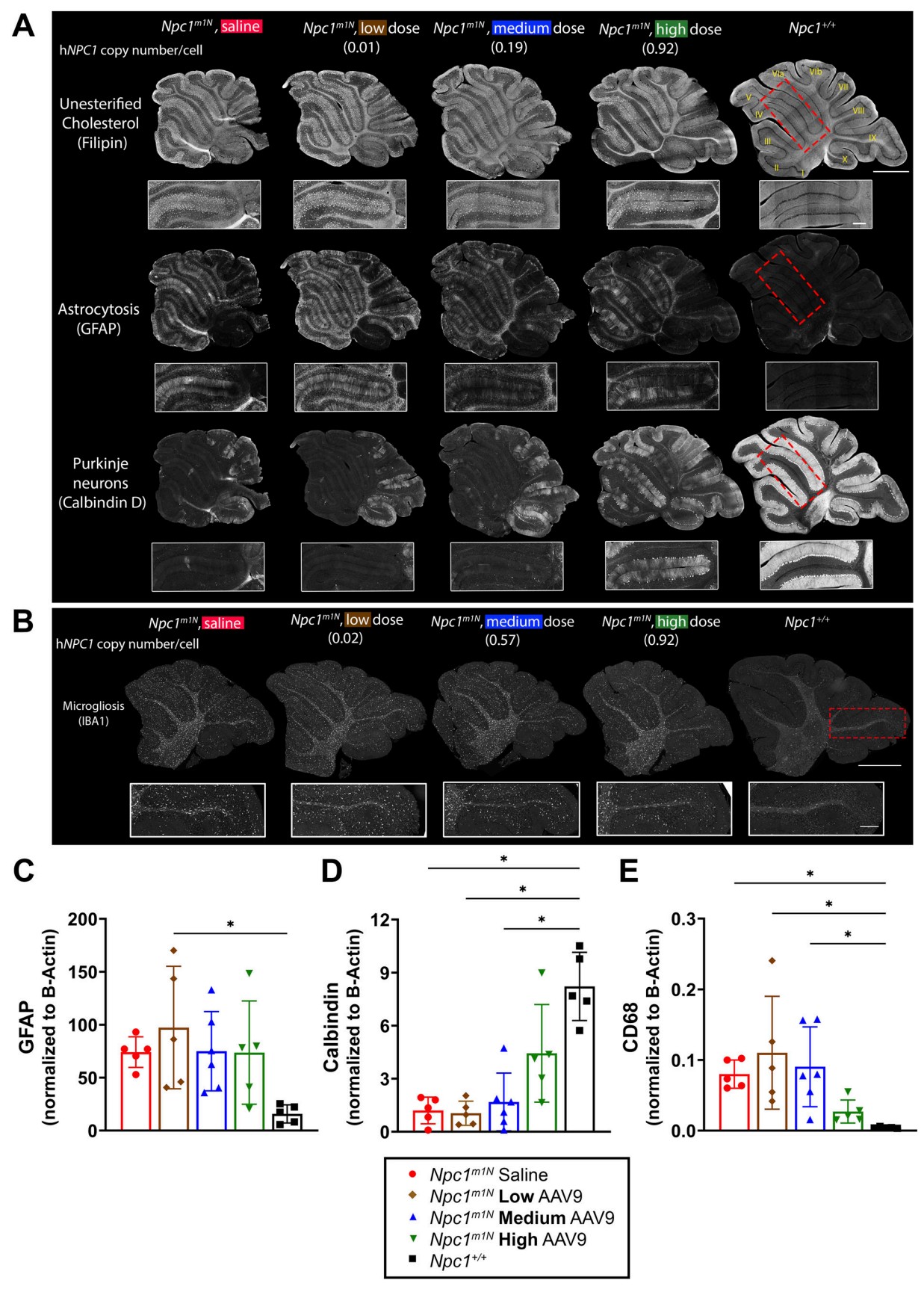

presymptomatic), 6 wk (early-stage disease), or 8 wk (late-stage disease) to assess the impact of treatment age on therapeutic efficacy. Disease onset is typically observed in *Npc1*$^{m1N}$ mice at 6 wk, with key manifestations including changes in motor coordination and ataxia that progress until death at ~10.6 wk (66). For this comparison, mice were euthanized at 9 wk to obtain age-matched cohorts or at humane endpoint for survival comparison.

Mice injected at 4 wk exhibited a significantly longer median survival of 21.5 wk compared with saline-injected mice (10.6 wk, *P* < 0.0001) and those treated at 6 wk (13.2 wk, *P* = 0.003) or 8 wk (11.9 wk, *P* < 0.0001) (Fig 5A). In addition, mice treated at 6 wk survived significantly longer than both saline-injected mice (*P* < 0.0001) and mice treated at 8 wk (*P* = 0.0003), whereas mice treated at 8 wk survived longer than saline-injected mice (*P* = 0.0006). Statistical analysis was performed using the log-rank Mantel–Cox test with Bonferroni's correction for multiple comparisons, with detailed results provided in Table S1B.

The composite phenotype was assessed between 6 and 18 wk to evaluate the effect of age of treatment on neurological disease progression (Fig 5B). Between 6 and 9 wk of age, mice treated at 4 wk old exhibited significantly lower phenotype scores compared with those at 6 and 8 wk (*P* = 0.001 and *P* < 0.0001, respectively), which progressed similarly (*P* = 0.51). From 9 to 12 wk, the 4-wk group maintained significantly lower composite scores than saline-injected mice (*P* < 0.0001), the 6-wk group (*P* = 0.0002), and the 8-wk group (*P* < 0.0001), but scores were higher than *Npc1*$^{+/+}$ mice (*P* < 0.0001). Two-way ANOVA with mixed-effects analysis and Tukey's multiple comparisons test was used for statistical analysis, with detailed results presented in Table S2B.

To assess disease onset and progression, we evaluated week of peak weight and percent weight change between 6 and 9 wk. Mice treated at 4 wk reached peak weight significantly later (11.3 ± 3.9 wk) than saline-injected mice (6.8 ± 0.7 wk, *P* = 0.003) and those treated at 6 wk (6.8 ± 2.3 wk, *P* < 0.001) or 8 wk (6.7 ± 0.6 wk, *P* < 0.001) (Fig 5C, Kruskal–Wallis with Dunn's multiple comparisons test). In contrast, mice treated at 6 and 8 wk reached peak weight at similar times as saline-injected mice (*P* = 0.82 and *P* > 0.99, respectively).

When evaluating percent weight change from 6 to 9 wk, mice injected at 4 wk (11.2% ± 13.1%) and *Npc1*$^{+/+}$ mice (6.0% ± 4.1%) exhibited weight gain and were significantly different from all other groups (*P* < 0.0001 for all comparisons). In contrast, *Npc1*$^{m1N}$ mice injected at 6 or 8 wk exhibited weight loss (−12.2% ± 7.7% or −20.0% ± 10.9%, respectively), which was not significantly different from saline-injected *Npc1*$^{m1N}$ mice (−14.6% ± 6.8%; *P* = 0.94 or *P* = 0.44, respectively) (Fig 5D, one-way ANOVA with Tukey's multiple comparisons test). Longitudinal weight data further indicate that both male and female cohorts treated earlier maintained weight and survived longer (Fig S1C and D; Table S3B).

## Age of treatment affects AAV9 transduction in the cerebellum, and treatment at 4 wk reduces cerebellar pathology in 9-wk-old *Npc1*$^{m1N}$ mice

The h*NPC1* copy number in the cerebellum was assessed in 9-wk-old mice. Mice injected at 4 and 6 wk had higher h*NPC1* copy numbers than saline mice or mice injected at 8 wk (Fig 6A). NPC1 protein levels were similar across all treated mice (Fig 6B; representative blot in Fig S6A and B).

Immunofluorescence staining and Western blots were used to evaluate cerebellar pathology in age-matched mice. Mice treated at 4 wk demonstrated greater Purkinje neuron survival in anterior lobules and the entire cerebellum when compared to saline-injected mice and those treated at 6 or 8 wk (Fig 6C); this pattern is also consistent with increased calbindin D protein levels in mice treated at 4 wk (Fig 6E; representative blot in Fig S6A and B). Furthermore, microgliosis appears mildly reduced in the posterior lobules of the cerebellum in the 4-wk group compared with saline-injected and other treated mice (IBA1 labeling, Fig 6D). A similar reduction was observed in CD68 protein levels in the 4-wk group, indicating decreased reactive microglial activity (Fig 6F; representative blot in Fig S6C and D). Finally, GFAP protein levels (reactive astrocytosis) appear similar across all groups (Fig 6G; representative blot in Fig S6C and D).

## Age of treatment impacts AAV9 transduction in the cerebrum and liver of *Npc1*$^{m1N}$ mice

The h*NPC1* copy number in the cerebrum and liver was also assessed via ddPCR in the age-matched cohort (9 wk) and at humane endpoint. In the age-matched cohort, mice injected at 8 wk exhibited the highest h*NPC1* copy number in the cerebrum compared with all other groups (Fig 7A). In the cerebrum, high copy numbers in 4- and 6-wk-old injected groups were associated with longer lifespans based on linear regression analysis; however, two mice with exceptionally high copy numbers in the 4-wk group drove significance in this finding. In mice treated at 8 wk, copy number did not predict lifespan (Fig 7B).

In the livers of the age-matched cohort, mice treated at 8 wk again had higher h*NPC1* copy numbers than other treated and saline-injected mice (Fig 7C). Linear regression analysis shows that higher copy numbers in the liver predicted lifespan only in mice treated at 6-wk old (Fig 7D). In mice from the age-matched cohort, immunohistochemical staining revealed a reduced presence of CD68$^+$ macrophages in the liver of mice treated at 6 wk compared with other treated and saline-injected mice (Fig 7E; representative images in Fig 7F).

---

**Figure 3.   Dose-dependent amelioration of cerebellar pathology in 10-wk-old *Npc1*$^{m1N}$ mice.**
**(A)** Representative immunofluorescence staining in free-floating sections for: unesterified cholesterol storage (top row), reactive astrocytes and Bergmann glia (middle row), and Purkinje neurons (bottom row). Insets are of anterior lobules (IV/V). **(B)** Representative immunofluorescence staining in formalin-fixed, paraffin-embedded sections for microgliosis. Insets are of posterior lobules (lobule IX). For (A, B), scale bar = 1,000 μm for panels, 250 μm for insets. **(C, D, E)** Protein levels for GFAP (C), calbindin (D), and CD68 (E) were assessed via Western blot for 10-wk-old mice (Kruskal–Wallis test with Dunn's multiple comparisons test) (saline, n = 5; low, n = 5; medium, n = 6; high, n = 5; *Npc1*$^{+/+}$, n = 5). For (C, D, E): **P* < 0.05, ***P* < 0.01, ****P* < 0.001, *****P* < 0.0001; data are presented as the mean ± SD.

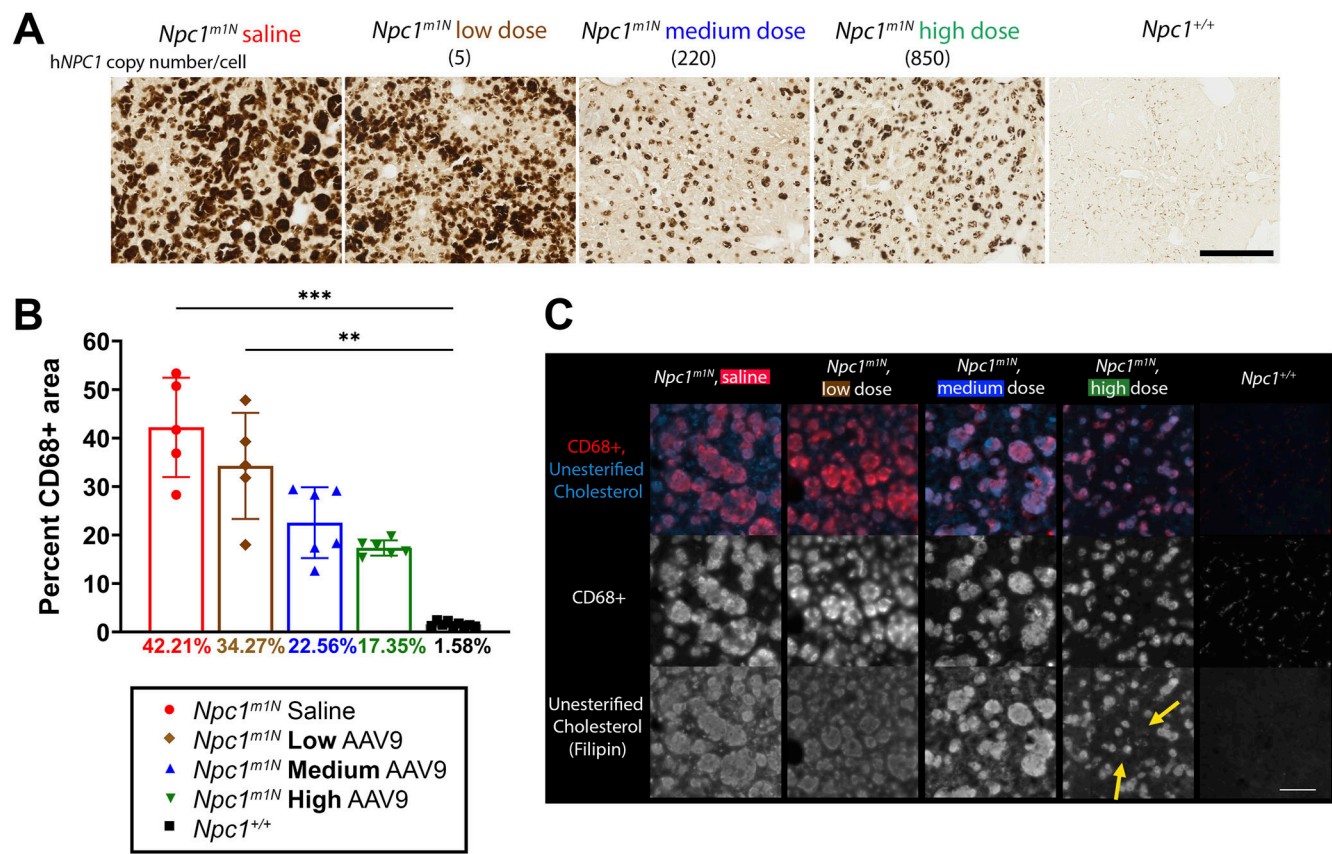

**Figure 4. Dose-dependent reduction of liver pathology in 10-wk-old $Npc1^{m1N}$ mice.**
**(A)** Representative CD68[+] immunohistochemical staining of macrophages (free-floating sections). Scale bar = 250 $\mu$m. **(B)** Quantification of percent area of CD68 labeled in 10-wk-old mice (Kruskal–Wallis test with Dunn's multiple comparisons test) (saline, n = 5; low, n = 5; medium, n = 6; high, n = 6; $Npc1^{+/+}$, n = 9). **(C)** Representative cholesterol storage (filipin labeling) and CD68[+] immunofluorescence staining in the liver (free-floating sections). Yellow arrows in high-dose inset denote groups of cells without cholesterol storage. **(A, C)** Scale bar for panel (A) = 500 $\mu$m, scale bar for (C) = 100 $\mu$m. For (B): *$P < 0.05$, **$P < 0.01$, ***$P < 0.001$, ****$P < 0.0001$; data are presented as the mean ± SD.

## $Npc1^{I1061T}$ mice treated with AAV9 show improved lifespan and delayed disease progression

To evaluate efficacy of AAV9 in a hypomorphic mouse model of NPC1 deficiency, $Npc1^{I1061T}$ mice were treated with 1.28 × 10$^{14}$ vg/kg (1.2 × 10$^{12}$ vg per mouse) of AAV9 at 4 wk old. In this substudy, mice were euthanized at 14 wk for an age-matched cohort or humane endpoint for survival.

AAV9-treated $Npc1^{I1061T}$ mice had a median survival of 22.9 wk, significantly longer than 15.0 wk observed in saline-injected mice (Fig 8A, log-rank Mantel–Cox test, $P < 0.0001$). Previous studies report a median survival of about 17.9 wk for $Npc1^{I1061T}$ mice, whereas $Npc1^{m1N}$ mice typically survive to 10.5 wk (65), and phenotypic onset in $Npc1^{I1061T}$ mice occurs at 9–10 wk; assessments were adjusted accordingly to account for differences in lifespan compared with the $Npc1^{m1N}$ model.

The disease phenotype was also evaluated in treated and untreated mice to determine efficacy of gene therapy in slowing progression (Fig 8B). From weeks 9 to 12 and weeks 12 to 15, AAV9-treated $Npc1^{I1061T}$ mice had significantly lower composite scores compared with saline-injected mice, but scores were still higher

than $Npc1^{+/+}$ mice ($P < 0.0001$ for all comparisons; two-way ANOVA with mixed-effects analysis and Tukey's multiple comparisons test).

Week of peak weight and percent weight change between 10 and 14 wk were evaluated to assess disease onset and progression. Treated mice reached peak weight significantly later than saline-injected mice (12.9 ± 3.8 wk versus 10.9 ± 1.1 wk; $P = 0.008$, Mann–Whitney test) (Fig 8C). Treated mice gained weight from 10 to 14 wk (3.8% ± 6.0%), similar to $Npc1^{+/+}$ mice (7.3% ± 2.7%) ($P = 0.37$). In contrast, saline-injected mice lost weight (−12.3% ± 11.0%) and were significantly different from treated mice ($P = 0.01$) (Fig 8D, Kruskal–Wallis test with Dunn's multiple comparisons test). Longitudinal weight data further demonstrate that treated mice survive and maintain weight longer than saline-injected $Npc1^{I1061T}$ mice (Fig S1E and F; Table S3C).

### AAV9 treatment transduces the cerebellum of $Npc1^{I1061T}$ mice but without pathological improvement

ddPCR was used to assess the h*NPC1* copy number in the cerebellum of mice at 14 wk old (age-matched cohort). At 14 wk, the

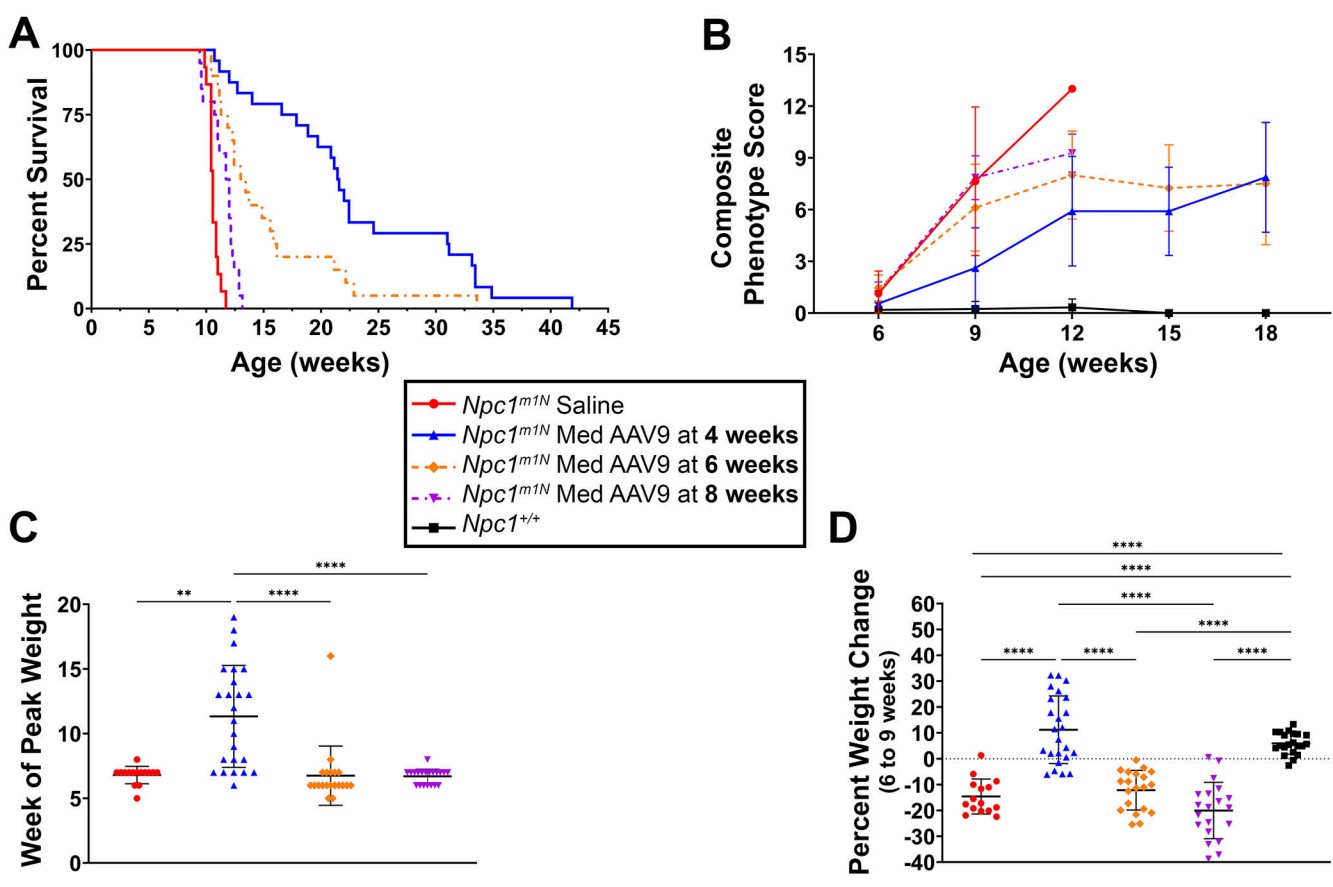

**Figure 5. AAV9 treatment at 4 wk improves survival and delays disease progression in *Npc1^{m1N}* compared with treatment at 6 or 8 wk of age.**
**(A)** Kaplan–Meier survival curve of mice treated with AAV9 (results presented in Table S1B). **(B)** Composite phenotype scores for each dosage group with measurements taken every 3 wk, starting at 6 wk (results presented in Table S2B) (saline, n = 14; 4 wk, n = 13; 6 wk, n = 20; 8 wk, n = 20; *Npc1^{+/+}*, n = 21). **(C)** Week at which mice reached peak weight (Kruskal–Wallis test with Dunn's multiple comparisons test). **(D)** Percent weight change between 6 and 9 wk old (one-way ANOVA with Tukey's multiple comparisons test). For (C, D): saline, n = 15; 4 wk, n = 24; 6 wk, n = 20; 8 wk, n = 20. For (D): *Npc1^{+/+}*, n = 21. For all: \*P < 0.05, \*\*P < 0.01, \*\*\*P < 0.001, \*\*\*\*P < 0.0001. **(B, C, D)** Data are presented as the mean ± SD for (B, C, D).

typical humane endpoint for untreated *Npc1^{I1061T}* mice, treated mice demonstrated increased h*NPC1* copy numbers compared with saline-injected mice (*P* = 0.008, Kruskal–Wallis test with Dunn's multiple comparisons test; Fig 9A). Parallel Western blot analysis confirmed similar NPC1 protein presence in treated and saline-injected *Npc1^{I1061T}* mice (NPC1 protein antibody captures both human and mouse NPC1) (Fig 9B; representative blot in Fig S7A and B).

Cerebellar pathology in the age-matched cohort was evaluated using immunofluorescence staining and Western blotting. *Npc1^{I1061T}* mice treated with AAV9 showed some survival of Purkinje neurons in anterior lobules of the cerebellum. However, global Purkinje neuron loss across the cerebellum remained comparable between treated and saline-injected mice (Fig 9C). Analysis of calbindin D protein levels revealed similarly low calbindin D levels in the cerebellum of treated and saline-injected mice, both of which were lower than healthy *Npc1^{+/+}* mice (Fig 9E; representative blot in Fig S7B and C). Microgliosis did not appear reduced in treated mice compared with saline-injected mice, as indicated by IBA1 labeling (Fig 9D). Similarly, CD68 protein levels, a reactive microglial marker, do not appear reduced in the cerebellum of

treated mice (Fig 9F; representative blot in Fig S7D and E). Finally, reactive astrocytosis, as measured by GFAP protein levels, is similar in treated mice compared with saline-injected mice (Fig 9G; representative blot in Fig S7D and E).

### AAV9 treatment of *Npc1^{I1061T}* mice results in successful transduction of the cerebrum and liver with reduced hepatic pathology

ddPCR was used to measure h*NPC1* copy number in the cerebrum and liver in the age-matched cohort (14-wk old) and at humane endpoint/survival. In the age-matched cohort, the h*NPC1* copy number in both the cerebrum (Fig 10A) and liver (Fig 10C) was elevated compared with saline-injected mice. In the cerebrum, an increasing h*NPC1* copy number significantly predicted extended lifespan (Fig 10B), whereas in the liver, an increase in h*NPC1* copy number was significantly associated with decreased lifespan (Fig 10D) (linear regression analysis). In the liver of age-matched mice, immunohistochemical staining revealed a partial reduction of CD68+ macrophages compared with saline-injected mice, albeit

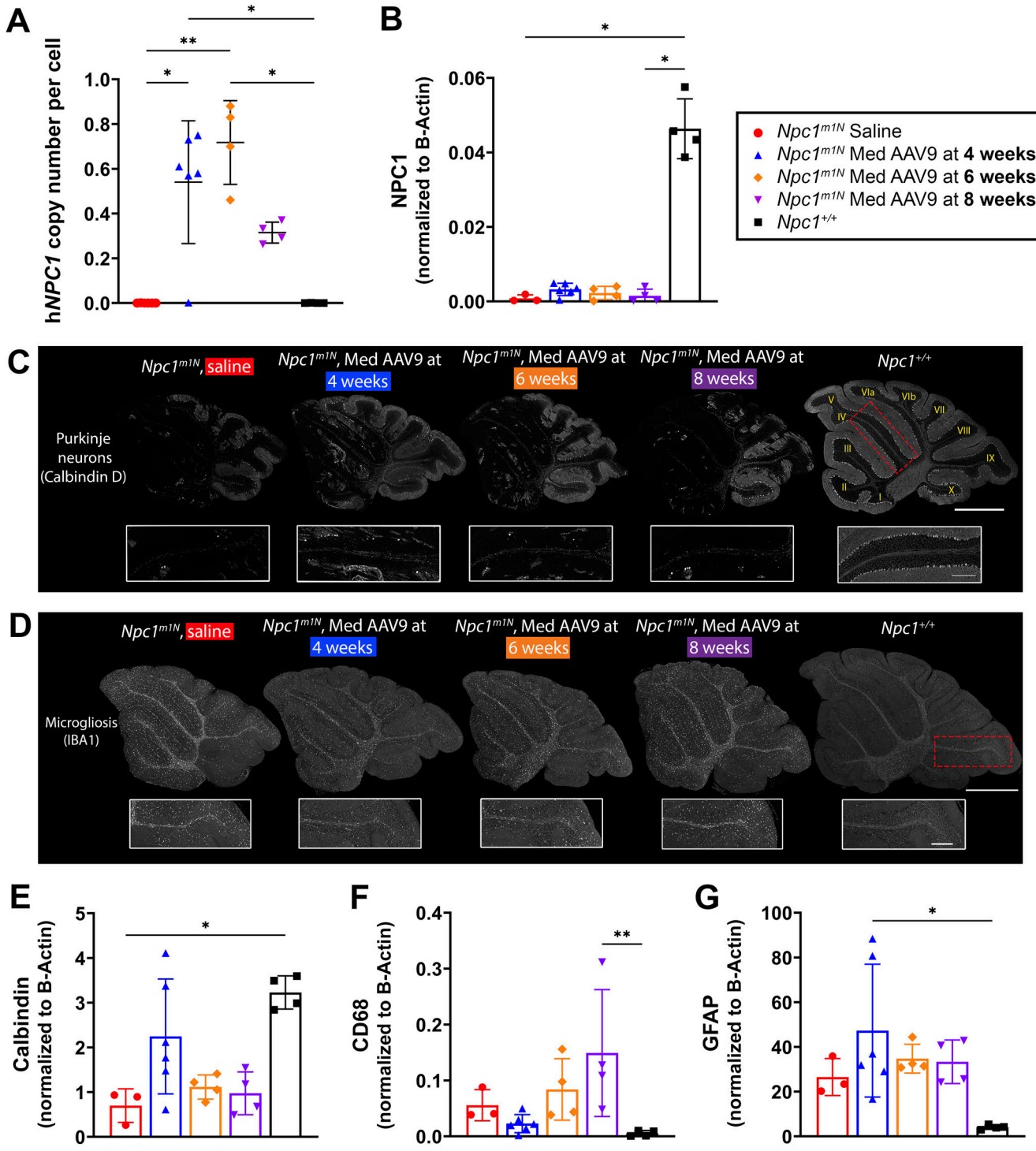

**Figure 6. Age of treatment affects AAV9 transduction in the cerebellum and treatment at 4 wk reduces cerebellar pathology.**
**(A)** ddPCR was used to measure h*NPC1* copy number in the cerebellum of mice at 9 wk old (Kruskal–Wallis test with Dunn's multiple comparisons test) (saline, n = 6; 4 wk, n = 6; 6 wk, n = 4; 8 wk, n = 4; *Npc1*[+/+], n = 4). **(B)** NPC1 protein levels were assessed via Western blot in 9-wk-old mice to confirm the amount of NPC1 protein in the cerebellum (Kruskal–Wallis test with Dunn's multiple comparisons test) (saline, n = 3; 4 wk, n = 6; 6 wk, n = 4; 8 wk, n = 4; *Npc1*[+/+], n = 4). **(C)** Representative immunofluorescence staining in a free-floating section for Purkinje neurons; insets are of anterior lobules (IV/V). **(D)** Representative immunofluorescence staining in formalin-fixed, paraffin-embedded sections for microgliosis. Insets are of posterior lobules (lobule IX). For (C, D): scale bar = 1,000 $\mu$m for panels, 250 $\mu$m for insets. **(E, F, G)** Protein levels for calbindin (E), CD68 (F), and GFAP (G) were assessed via Western blot for each 9-wk-old mouse cohort (Kruskal–Wallis test with Dunn's multiple comparisons test) (saline, n = 3; 4 wk, n = 3; 6 wk, n = 4; 8 wk, n = 4; *Npc1*[+/+], n = 4). For all: *$P < 0.05$, **$P < 0.01$, ***$P < 0.001$, ****$P < 0.0001$. **(A, B, E, F, G)** Data are presented as the mean ± SD for (A, B, E, F, G).

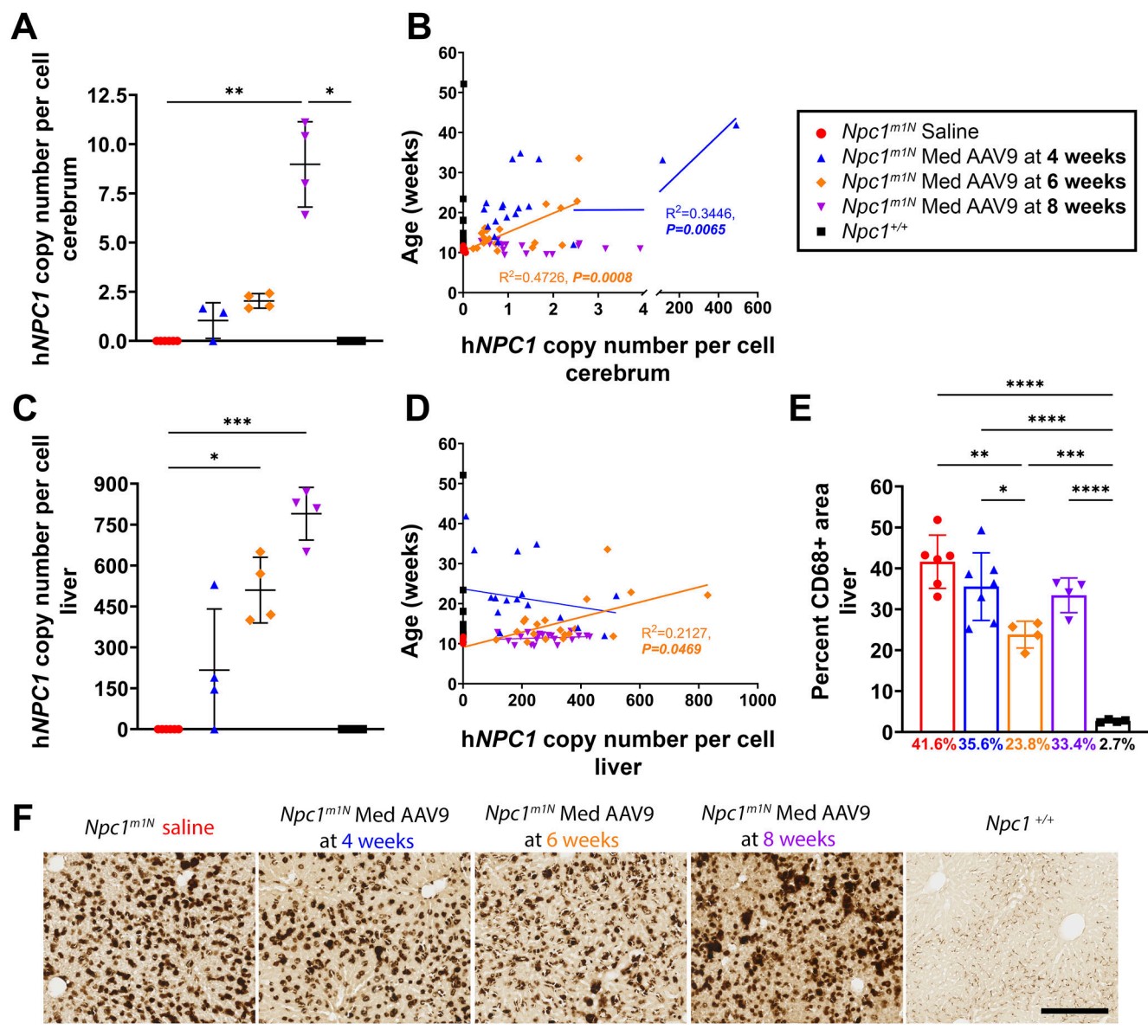

**Figure 7. Age of treatment impacts AAV9 transduction in the cerebrum and liver as well as hepatic pathology.**
**(A, C)** ddPCR was used to measure h*NPC1* copy number in mice at 9 wk old in the cerebrum (A) and liver (C) (Kruskal–Wallis test with Dunn's multiple comparisons test). **(C)** For (A): saline, n = 6; 4 wk, n = 3; 6 wk, n = 4; 8 wk, n = 4; *Npc1*[+/+], n = 4. For (C): saline, n = 6; 4 wk, n = 4; 6 wk, n = 4; 8 wk, n = 4; *Npc1*[+/+], n = 4. **(B, D)** Linear regression of lifespan as a function of h*NPC1* copy number in the cerebrum (B) and liver (D). For (B): saline, n = 14; 4 wk, n = 19; 6 wk, n = 20; 8 wk, n = 20; *Npc1*[+/+], n = 17. For (D): saline, n = 14; 4 wk, n = 18; 6 wk, n = 20; 8 wk, n = 20; *Npc1*[+/+], n = 17. **(E)** Quantification of percent area of CD68 labeled in 9-wk-old mice (Kruskal–Wallis test with Dunn's multiple comparisons test) (saline, n = 6; 4 wk, n = 8; 6 wk, n = 4; 8 wk, n = 4; *Npc1*[+/+], n = 4). **(F)** Representative immunohistochemical staining of macrophages in the liver in 9-wk-old mice (free-floating sections). Scale bar = 250 μm. For all: *$P < 0.05$, **$P < 0.01$, ***$P < 0.001$, ****$P < 0.0001$. **(A, C, E)** Data are presented as the mean ± SD for (A, C, E).

---

not to normal *Npc1*[+/+] levels (Fig 10E; representative images in Fig 10F).

## Discussion

We examined dose and age at intervention as key factors in optimizing systemically delivered AAV9-EF1a(s)-h*NPC1* gene therapy. Higher doses of AAV9-EF1a(s)-h*NPC1* and treatment during the presymptomatic period in the null *Npc1*[m1N] model significantly improved survival and slowed disease progression compared with lower doses or later treatment. Notably, mice receiving higher doses had the longest survival times, though they did ultimately succumb to NPC1 disease. In addition, we assessed efficacy of gene therapy in the hypomorphic *Npc1*[I1061T] mouse, which carries a missense variant, and found that AAV9-EF1a(s)-h*NPC1* successfully increased survival and slowed disease progression in this model.

NPC1 can manifest early and severely, though many individuals experience the first neurological sign in childhood or adolescence,

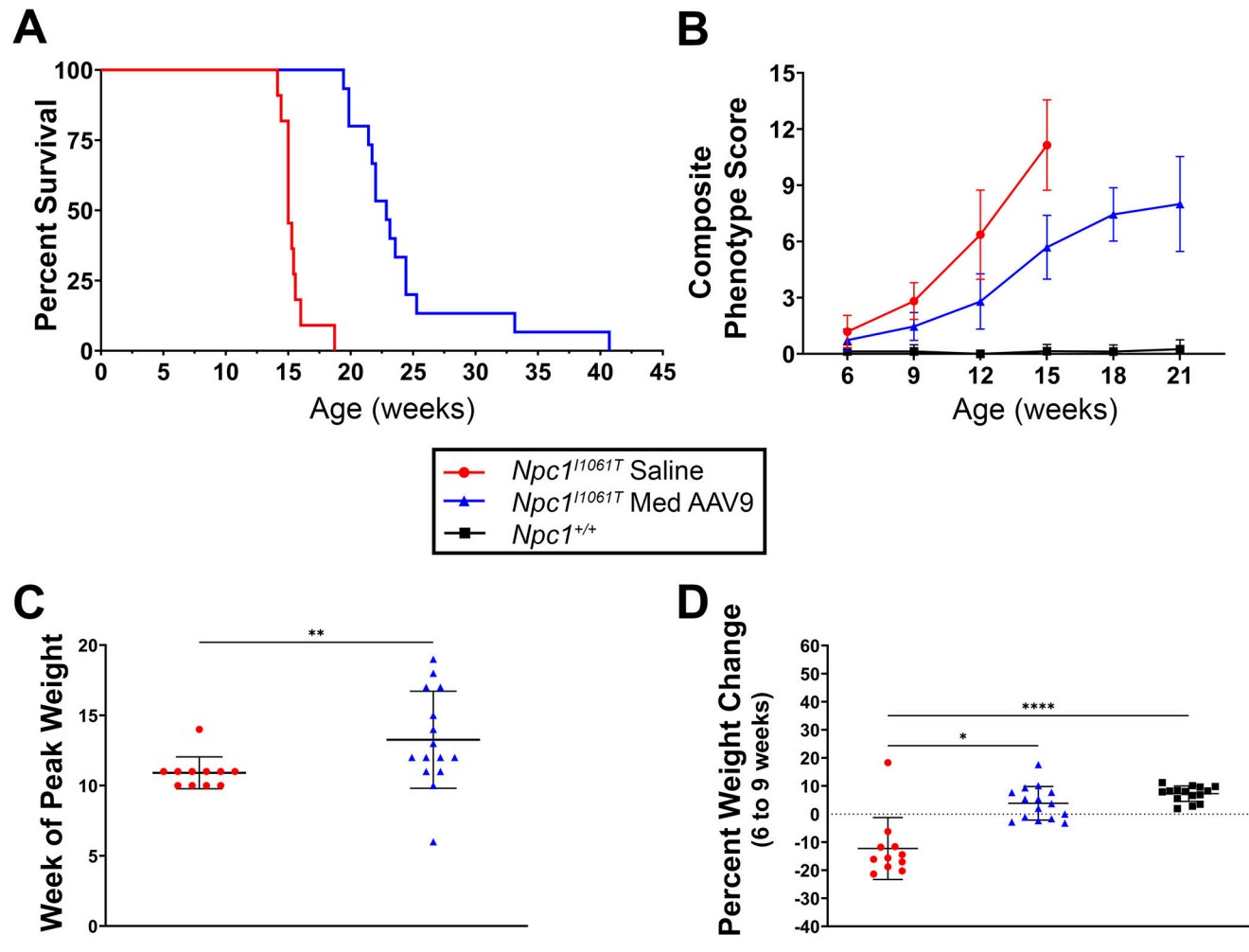

**Figure 8.** *Npc1*[I1061T] **mice treated with AAV9 show increased lifespan and delayed disease progression.**
**(A)** Kaplan–Meier curve depicts survival of saline-injected *Npc1*[I1061T] mice and *Npc1*[I1061T] mice treated with AAV9 at 4 wk. **(B)** Composite phenotype score for each group was measured from 6 to 21 wk of age at 3-wk intervals. **(C)** Week mice reached peak weight (Mann–Whitney test). **(D)** Percent weight change from 10 to 14 wk (Kruskal–Wallis with Dunn's multiple comparisons). For (A, B, C, D): saline, n = 11; treated, n = 15; and for (B, D): *Npc1*[+/+], n = 15. For all: *$P < 0.05$, **$P < 0.01$, ***$P < 0.001$, ****$P < 0.0001$. For (B, C, D): data are presented as the mean ± SD.

often followed by significant diagnostic delays. Previous gene therapy studies have focused on treatment of neonatal *Npc1*[m1N] mice (42, 43, 45, 46). Importantly, we assessed gene therapy efficacy in juvenile mice before and after symptom onset. *Npc1*[m1N] mice treated presymptomatically showed greatest improvements in survival and slowed disease progression. However, mice treated in early disease stage (6 wk) still exhibited slight survival benefits, indicating the value of early-symptomatic treatment. Establishing a dose- and age-dependent effect is crucial, especially for diseases like NPC1, which are typically diagnosed symptomatically and are not currently part of newborn screening. By demonstrating the benefits of early intervention, our work highlights the need for expanded newborn screening programs, which would enable earlier diagnosis and support treatment of NPC1 individuals before neurological manifestations occur.

Although the *Npc1*[I1061T] model has not been previously studied, another hypomorphic mouse model (*Npc1*[nmf164]), which carries a D1005G amino acid substitution, has been examined in gene

therapy studies (46). Both models exhibit late-onset and slower progression compared with *Npc1*[m1N] mice, which more closely mirrors most human cases. However, the D1005G residue is not conserved in humans and its impact on protein folding or stability is unclear (81). In contrast, the *Npc1*[I1061T] model represents a prevalent human variant, with NPC1 p.I1061T encoding a misfolded protein targeted for ERAD (64, 65). This variant is the most common disease-causing allele in individuals of European descent, accounting for 15–20% of pathological alleles (62, 82, 83). We demonstrate significant improvement in survival, delayed disease progression, and liver (though not cerebellar) pathology with systemic gene therapy, even in the presence of the residual NPC1 protein. These findings might suggest broader applicability of gene therapy across different patient populations, including those with slower progressing, later onset NPC1.

Although our high dose showed the most significant survival benefits and improved transduction of multiple organs affected by NPC1, its direct translation to humans requires careful

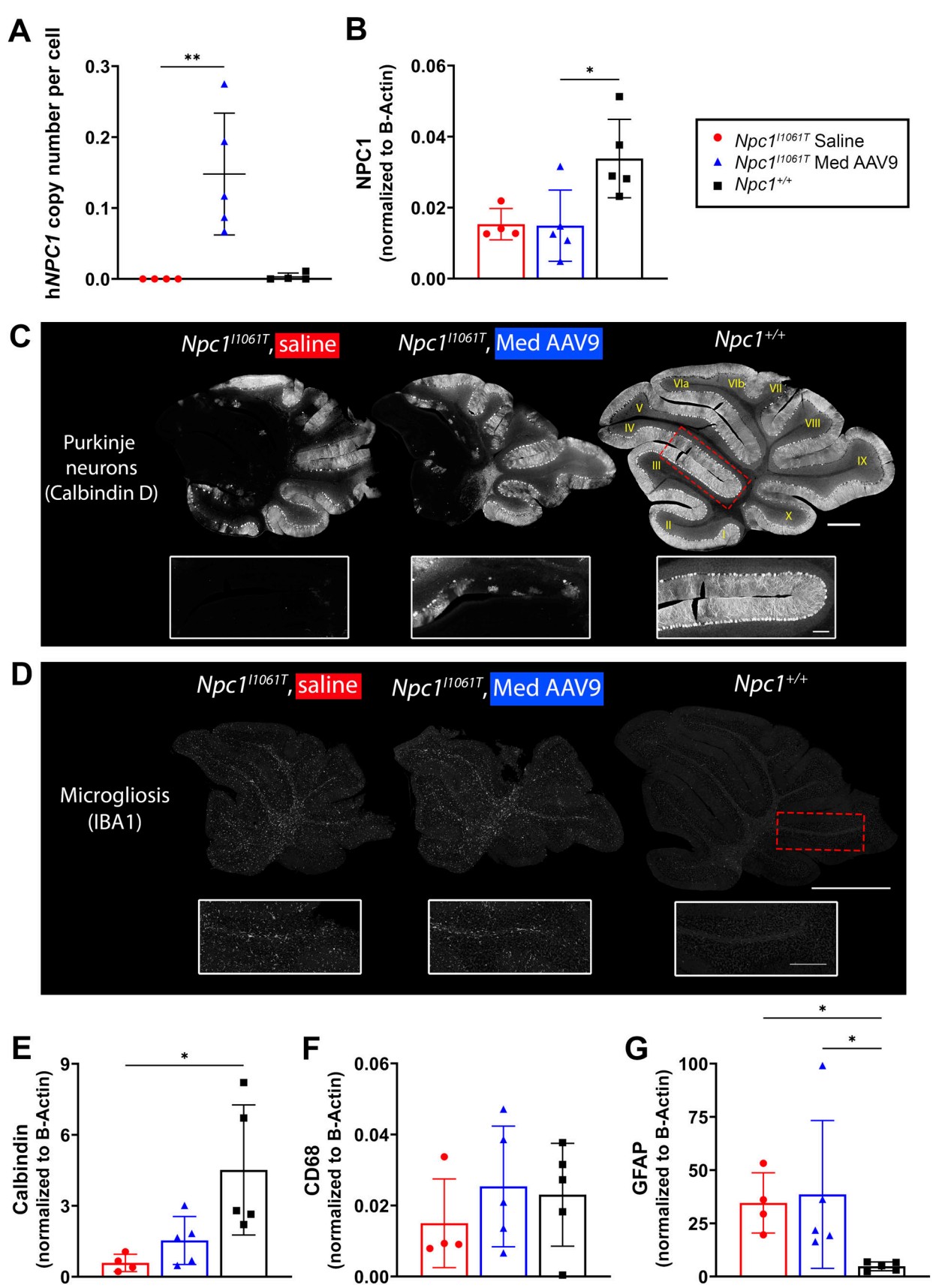

consideration. For comparison, Zolgensma, an FDA-approved therapy for SMA1, is administered at $1.1 \times 10^{14}$ vg/kg, whereas our high dose of $3.06 \times 10^{14}$ vg/kg exceeds that level. Although we did not observe toxicity at this higher dose, clinical experiences with AAV8-mediated gene therapy for X-linked myotubular myopathy underscore potential risks, where 3 of 14 patients experienced fatal liver failure at a dose of $3.5 \times 10^{14}$ vg/kg (84). These observations highlight the importance of dose escalation studies in preclinical models to refine dosing strategies for translation to human applications. Our medium AAV9 dose of $1.28 \times 10^{14}$ vg/kg—comparable to Zolgensma—showed significant improvements in survival and disease progression, suggesting that doses within this range could balance efficacy and safety. Mice receiving higher doses had the longest survival times, though they ultimately succumbed to disease progression. As necropsies were not carried out, the exact cause of death at the predefined humane endpoint is not known. We hypothesize that because many neurons and cells throughout the CNS remain uncorrected after a single injection of the gene therapy, the disease burden eventually outweighs the capabilities of the transduced, healthy cells. If correct, this theory highlights the importance of widespread transduction, best-case scenario being correction of 100% of cells, especially of neurons throughout the CNS. Recognizing this is not yet possible, toxicity must be assessed for high doses aimed at maximizing transduction before clinical translation. Together, these findings support the notion that increasing doses achieve greater benefit, the importance of achieving adequate dosing for therapeutic efficacy, and the need to balance efficacy with safety in future studies.

Regarding readout measures of mice in the same-dose group, a comparison of weight metrics reveals greater variability in the medium- and high-dose treatment groups, whereas survival in the medium-dose group covers a wide range. Retro-orbital administration, as performed by human hand, is subject to inconsistency. To minimize this potential confounder, a veterinarian with extensive surgical experience in rodent models was the only person to perform RO injections for gene therapy studies from our group to date. Another potential contributor to variability within the different readout measures is weight at time of injection. However, no significant correlation was found between this data metric and the survival, h*NPC1* copy number, or percent weight gain from 6 to 9 wk of age. Finally, handling of vector and different vector productions could contribute to variability. To minimize this risk, all vector productions were purchased from University of Pennsylvania Vector Core, now Franklin Biolabs. Vector was thawed only once after dilution, and the same vector production batch was used for the high- and low-dose cohorts. The medium-dose cohort/study spanned a greater range of time and provided an internal control as three different batches were used. There was no correlation between the vector batch and lifespan of the mice in the medium-dose cohort.

To compare the effects of age of treatment in *Npc1^{m1N}* mice, we used a fixed-analysis point at 9 wk of age, just before humane endpoint. This age-matched time point allowed for consistent comparisons of pathology and vector transduction but did not account for treatment duration–matched intervals (e.g., assessing all mice at a fixed period, such as 2 wk posttreatment). Notably, mice treated at 8 wk exhibited higher h*NPC1* copy numbers in the cerebrum and liver compared with those treated at 4 or 6 wk. However, it is unclear whether this trend would persist in a treatment duration–matched cohort. Future studies could explore posttreatment intervals to better understand how vector expression changes over time and more directly compare the impact of age of treatment on transduction efficiency.

In addition, we did not determine which cell types were transduced by AAV9 administration or whether neuronal dysfunction or loss contributed more to pathology. Future studies might assess the transduction of cell types across the brain, investigating both cell death and dysfunction. For instance, we observed Purkinje neuron loss in the cerebellum (a well-described phenomenon (67)), but others have described lipid storage without cell loss in the pyramidal cells of the hippocampus (85, 86). Understanding which cell types are most affected and best transduced is essential for clinical success of this gene therapy. Another limitation of this study is the absence of direct quantification of accumulating metabolites in the brain. For example, although the overall cholesterol levels in the brain remain unchanged in NPC1, its intracellular localization is disrupted, leading to pathological effects (8, 87). To address lipid accumulation, we used mass spectrometry imaging to measure changes in sphingolipids that change in both quantity and distribution. This provided valuable insights into lipid changes in response to treatment, but future studies could incorporate high-performance liquid chromatography to enhance our understanding of lipid storage correction after gene therapy by broadening the analysis of lipid metabolites.

Future studies could also explore combining NPC1 gene therapy with chaperone molecules or substrate reduction therapy, given that gene therapy alone does not entirely halt disease progression. For example, miglustat, an FDA-designated compound for use in combination with arimoclomol (Miplyffa), reduces glycosphingolipid accumulation but does not address the root cause (10, 88, 89). Incorporating gene therapy could address this root cause while enhancing therapeutic outcomes by simultaneously reducing disease burden.

In summary, systemic AAV9-EF1a(s)-h*NPC1* delivery significantly impacts NPC1 disease phenotypes and improves survival in both

**Figure 9. AAV9 treatment effectively transduces the cerebellum of *Npc1^{I1061T}* mice, and modestly impacts cerebellar pathology.**
**(A)** ddPCR was used to measure h*NPC1* copy number in the cerebellum in mice at 14 wk old (Kruskal–Wallis test with Dunn's multiple comparisons test) (saline, n = 4; AAV9, n = 5; *Npc1^{+/+}*, n = 4). **(B)** NPC1 protein levels were assessed via Western blot in 14-wk-old mice to determine the amount of NPC1 protein in the cerebellum (Kruskal–Wallis test with Dunn's multiple comparisons test) (saline, n = 4; AAV9, n = 5; *Npc1^{+/+}*, n = 5). **(C)** Representative immunofluorescence staining in a free-floating section for Purkinje neurons; insets are of anterior lobules (IV/V). **(D)** Representative immunofluorescence staining in formalin-fixed, paraffin-embedded sections for microgliosis. Insets are of posterior lobules (lobule IX). For (C, D): scale bar = 1,000 $\mu$m for panels, 250 $\mu$m for insets. **(E, F, G)** Protein levels for calbindin (E), CD68 (F), and GFAP (G) were assessed via Western blot for each 9-wk-old mouse cohort (Kruskal–Wallis test with Dunn's multiple comparisons test) (saline, n = 4; AAV9, n = 5; *Npc1^{+/+}*, n = 5). For all: *$P < 0.05$, **$P < 0.01$, ***$P < 0.001$, ****$P < 0.0001$. **(A, B, E, F, G)** Data are presented as the mean ± SD for (A, B, E, F, G).

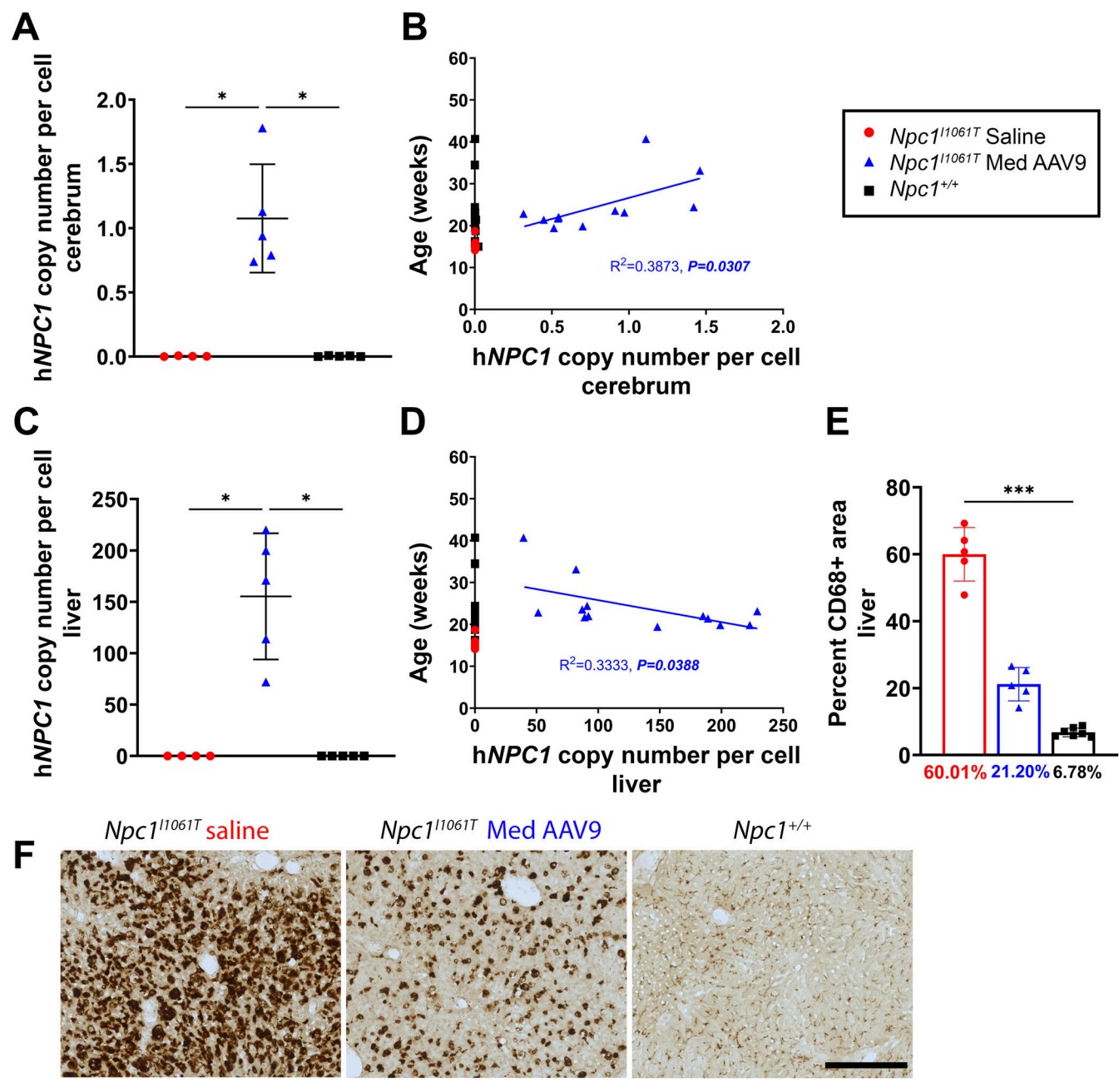

**Figure 10. Transduction efficacy of AAV9 in *Npc1^I1061T* mouse cerebrum and liver tissue and hepatic pathology.**
**(A, C)** ddPCR was used to measure h*NPC1* copy number at 14 wk in the cerebrum (A) and liver (C). For (A, C): saline, n = 4; treated, n = 5; *Npc1^+/+*, n = 5 (Kruskal–Wallis test with Dunn's multiple comparisons test). **(B, D)** Linear regression of lifespan as a function of h*NPC1* copy number in the cerebrum (B) or liver (D). For (B): saline, n = 11; treated, n = 12; *Npc1^+/+*, n = 14. **(D)**: saline, n = 11; treated, n = 13; *Npc1^+/+*, n = 14. **(E)** Quantification of percent area of CD68 labeled in 14-wk-old mice (Kruskal–Wallis test with Dunn's multiple comparisons test) (saline, n = 5; treated, n = 5; *Npc1^+/+*, n = 6). **(F)** Representative immunohistochemical staining of macrophages in the liver of saline-injected, treated, and *Npc1^+/+* mice at 14 wk (free-floating sections). Scale bar = 250 $\mu$m. For all: *$P < 0.05$, **$P < 0.01$, ***$P < 0.001$, ****$P < 0.0001$. For (A, C, F): data are presented as the mean ± SD.

severe and milder disease models, with delivery of high doses and presymptomatic treatment providing the greatest impact. The findings presented here lay the groundwork for translating this promising therapy to clinical trials for individuals with NPC1 deficiency. Our work has broader implications for gene therapy targeting other lysosomal membrane–spanning proteins, such as those implicated in neuronal ceroid lipofuscinosis (20, 90, 91) and mucolipidosis type IV (92, 93). The ability of gene therapy to improve survival and delay disease progression in juvenile mice offers hope for diseases where early detection and intervention remain significant challenges. Although gene therapy may not entirely halt disease progression in NPC1, the demonstrated

survival benefits emphasize its potential in managing other similar disorders where early, aggressive intervention is critical.

# Materials and Methods

### Vector construction and production

The vector AAV9-EF1a(s)-h*NPC1* was previously described and produced by the University of Pennsylvania Vector Core, now Franklin Biolabs (18).

### Animals

All animal work in these studies was carried out in accordance with the National Institutes of Health Animal Care and Use Committee–approved protocols. Heterozygous (BALB/cNctr-*Npc1^{m1N}*/J strain; strain # 003092; Jackson Laboratory) *Npc1^{+/m1N}* mice were crossed to obtain homozygous mutants (*Npc1^{m1N/m1N}*) and wild-type controls (*Npc1^{+/+}*). *Npc1^{I1061T}* mice were generated by crossing heterozygous *Npc1^{+/I1061T}* mice (B6.129-*Npc1^{tm1Dso}*/J strain; strain # 027704; Jackson Laboratory) to obtain homozygous mutants and *Npc1^{+/+}* controls. We use single allele notation to indicate homozygosity. Mice were weighed weekly and then more frequently as the disease progression neared humane endpoint. Mice were euthanized at a predefined humane endpoint, which occurred when at least two of the following four criteria were met: weight falling below 70% of peak weight, repeatedly falling to side during movement, dull eyes or palpebral closure of eyes, or reluctance to move.

As per the ARRIVE Essential 10, the following information refers to the studies contained herein. All studies included saline-injected or untreated mice (*Npc1^{m1N}* or *Npc1^{+/+}*, respectively) as control groups. The same control mice from the *Npc1^{m1N}* line were used for the dose and age at injection studies, whereas appropriate groups of control mice from the *Npc1^{I1061T}* line were used for the corresponding study. A total of 238 mice were used in these studies. Group sample size is stated in the legend or figure for each analysis. Data for dose and age of treatment study include all saline-injected and medium-dose AAV9 *Npc1^{m1N}* mice, as well as untreated *Npc1^{+/+}* at 4 wk. No specific exclusion criteria were set a priori, and all mice included in the studies were randomly assigned to treatment or control groups using a blocking method. Except for the researcher overseeing the studies, those involved in vector injections and data acquisition remained blinded to the greatest extent possible. The primary outcome measure was survival. Secondary outcome measures included behavioral assessments, weight, h*NPC1* copy number, and pathology. Details of statistical analyses are found in the results, methods, or figure legends. Experimental animals, procedures, and results are contained within the methods and results of this publication.

### Phenotypic assessment

Mice were tested starting at 6 wk of age (phenotypic onset) and every 3 wk thereafter until humane endpoint or inability to complete the evaluation. The phenotype score evaluates five behaviors associated with the NPC1 phenotype in diseased mice as previously described: hindlimb clasp, motor function, kyphosis, grooming, and a balance-ledge test for cerebellar ataxia (66). Each phenotype is scored from 0 to 3 with increasing scores representing a more compromised disease state. Briefly, mice are scored as follows. Hindlimb clasp (holding mouse by base of tail in air for 10 s): 0 = hindlimbs consistently splayed outward away from abdomen; 1 = hindlimbs individually retract toward abdomen for cumulative time of 1–4 s; 2 = hindlimbs individually retract toward abdomen for cumulative time of >/= 5 s; and 3 = both hindlimbs retract toward abdomen for any time. Motor function (freely moving about empty cage): 0 = normal movement, abdomen not touching ground, all limbs support body weight, mouse able to rear; 1 = slightly uneven gait with possible tremor or limp, mouse able to rear but hindlimbs often slide back; 2 = splayed hindlimbs when locomoting, reluctance to rear with little upward extension; and 3 = difficulty moving forward, abdomen touches ground, generally unable/unwilling to rear. Kyphosis (curvature of spine observed while at rest and during locomotion): 0 = straight spine during locomotion, no overt kyphosis at rest; 1 = mild kyphosis at rest but able to straighten spine while moving; 2 = persistent but mild kyphosis while stationary and while in motion; and 3 = obvious, pronounced kyphosis and unable to straighten spine. Grooming (mouse observed in empty cage): 0 = normal coat with smooth, shiny fur; 1 = mostly normal coat with some piloerection; 2 = patchy, ungroomed coat on upper half of body; and 3 = patchy, ungroomed coat on most or all of dorsal body. Balance-ledge test (mouse observed while traversing cage ledge): 0 = maintains balance and hindlimbs remain on top of ledge; 1 = hindlimbs mostly on top of ledge but less coordinated and some hindlimb slips while traversing; 2 = hindlimbs not effectively used to navigate ledge and hug the sides rather than remain on top of ledge; and 3 = mouse unable to navigate ledge, often falls off or refuses to move.

Phenotypic testing was carried out in a blinded fashion such that individual mice within a cage had distinct tail markings. Evaluators had access only to cage card numbers and tail markings to identify mice. The order in which mice were tested varied for each testing date. All animals were group-housed.

### Administration of vector

Treated *Npc1^{m1N}* mice received a RO injection of AAV9-EF1a(s)-h*NPC1* at 4 wk (weaning), 6 wk, or 8 wk of age. *Npc1^{I1061T}* mice received a RO injection of AAV9-EF1a(s)-h*NPC1* at 4 wk old. Control littermate *Npc1^{m1N}* or *Npc1^{I1061T}* mice received a RO injection of 0.9% saline at 4 wk or the specified age. Mice were anesthetized using isoflurane for 30–60 s and then injected RO with a 30-gauge needle affixed to a 0.3 cc syringe. Some control *Npc1^{+/+}* mice received RO injections of saline, whereas others remained uninjected.

### Dose selection

Our original dose for mice treated at 4 wk was $1.2 \times 10^{12}$ vector genomes (vg) per mouse (mean: $1.28 \times 10^{14}$ vg/kg), based on our previously published data (18, 44). This served as the medium dose

and the comparator across studies. For ease of comparison with other animal and clinical studies, doses are provided as vector genomes/kg.

For dose comparison, doses were selected and scaled based on vector genomes/mouse. We used a low dose of $1 \times 10^{11}$ vg/mouse ($7.87 \times 10^{12}$ vg/kg), representing a 1-log reduction from the baseline. The high dose, $4.3 \times 10^{12}$ vg/mouse ($3.06 \times 10^{14}$ vg/kg), was dictated by the maximum concentration achievable through the University of Pennsylvania Vector Core, now Franklin Biolabs, and volume constraints allowed by the Animal Care and Use Committee. For the age of treatment study, dose was calculated per weight to deliver $1.28 \times 10^{14}$ vg/kg for each mouse at either 6 or 8 wk of age.

### Tissue collection and homogenization

Mice were anesthetized with an intraperitoneal injection of avertin (lethal dose of 0.04 ml/gm) for euthanasia as previously described (44). When mice were insensate, the chest cavity was opened, and mice were perfused with 0.9% saline. Immediately, half of the brain, one lobe of the liver, and a piece of spleen, kidney, lung, and leg muscle were collected and frozen on dry ice for tissue homogenization. Mice were then perfused again with 4% paraformaldehyde to fix tissues; remaining organs (half of the brain, liver) were collected and stored postfixation in 4% PFA overnight and then rinsed and stored in PBS.

A Benchmark Scientific BeadBug homogenizer was used to homogenize frozen tissue with UltraPure water. Tissue was placed in tubes with 3-mm zirconium beads (cerebrum, cerebellum, brainstem, liver, spleen, leg muscle) or 1.5-mm zirconium beads (kidney, lung) and homogenized three times for 30 s at speed 400. The resulting homogenate was aliquoted into tubes for DNA extraction and protein analysis, the latter of which also had RIPA buffer with proteinase inhibitor cocktail (11 836 170 001; MilliporeSigma) (44).

### Western blotting

Protein levels from cerebrum, cerebellum, and liver homogenates were quantified using a BCA assay (23225; Thermo Fisher Scientific). Equal amounts of protein (50 $\mu$g for liver, cerebellum, and cerebrum) were run on 4–12% Bis-Tris SDS–polyacrylamide gels (NP0321BOX/Invitrogen by Thermo Fisher Scientific), and separation was achieved via electrophoresis; protein was then transferred to a nitrocellulose membrane (Life Technologies) and blocked for 1 h in 5% milk in 0.01% PBS/Tween. Samples were washed in 0.01% PBS/Tween incubated overnight at 4°C on a rocking platform with primary antibody (Table S4). Blots were washed three times and incubated with secondary antibodies for 1 h at RT (Table S4), then washed again. Bands were imaged using the LI-COR Odyssey CLx Imaging System.

### Histology

Brain and liver tissues from each group were acquired at 9 or 10 wk of age ($Npc1^{m1N}$), 14 wk of age ($Npc1^{I1061T}$), or humane endpoint (both models). Postfixation tissues were embedded in agarose blocks (3.5% agarose, 8% sucrose, PBS) and sectioned

parasagittally (30 $\mu$m) using a Leica VT1200 S vibratome. Free-floating sections were collected, incubated in 1.6% $H_2O_2$ in PBS, then washed in 0.25% Triton X-100/PBS (PBSt). After blocking for 1 h at RT in PBSt/normal goat serum, samples were incubated in primary antibodies overnight at 4°C (Table S4). Samples were washed in PBSt and then incubated with secondary antibodies for 1 h at RT (Alexa Fluor 488 or 594; Table S4). Filipin staining (F9765; Sigma-Aldrich) was finally performed to allow visualization of unesterified cholesterol accumulation (0.05 mg/ml) with a 20-min incubation. ProLong Gold mounting medium alone (P36934; Thermo Fisher Scientific) or with DAPI (P36935; Thermo Fisher Scientific) was used to coverslip after mounting tissue sections to slides.

For immunohistochemical staining after primary antibody incubation, slides were incubated in biotinylated secondary antibody and washed in PBSt. A biotinylated horseradish peroxidase was preincubated with avidin to form avidin–biotin complex (ABCSK-4100; Vector Laboratories), and the tissues were incubated in ABC (in PBS) for 1 h. Tissues were then washed in PBS and incubated for 10 min in a 3,3′-diaminobenzidine (DAB,PK-4000; Vector Laboratories) solution before mounting and coverslipping with VectaMount (H-5700; Vector Laboratories).

Histoserv, Inc. performed paraffin embedding (formalin-fixed, paraffin-embedded, i.e., FFPE, tissues). For immunofluorescence staining, FFPE sections (3 $\mu$m) were collected and underwent antigen retrieval in a citrate (pH 6.0, 62706-10; Electron Microscopy Sciences) or Tris-EDTA (pH 9.0, AB93684; Abcam) buffer. Slides were then incubated in primary antibody diluted in antibody diluent with BSA and preservative (003218; Thermo Fisher Scientific) at 37°C for 1 h, washed in PBS, and incubated in secondary antibodies for 30 min at 37°C (Table S4). Tissues were then coverslipped with ProLong Gold mounting medium with DAPI.

### Image capture and analysis

Immunofluorescence and bright-field imaging was performed with an inverted Zeiss AxioScan.Z1 slide scanner using a 20× objective with Zen Blue 3.8 as previously described in reference 44. All tissues for each combination of antibodies were stained in the same run, imaged using the same acquisition parameters, and processed identically to ensure accuracy of qualitative and quantitative comparisons. Adobe Photoshop 2023 (v.23.5.0) and 2024 (v.25.1.0) were used to modify all images in a figure/group identically by resizing and adjusting brightness and/or contrast.

### Quantification of the CD68 area

The percentage of positive CD68 area relative to total area in liver sections was determined according to methods previously described (94) using Image-Pro v11 software (Media Cybernetics, Inc.). Briefly, 10 regions of interest (ROI) were selected from two to three tissue sections per mouse for a total area of 900,000 $\mu$m$^2$. Using the same threshold for all tissues within a comparative analysis, Image-Pro software outlined the stained areas (i.e., CD68$^+$) within each ROI and this value was then divided by the

total area to obtain the percent positive CD68 area for each mouse liver.

## Copy-number analysis by ddPCR

Gene copy number was quantified using the QX200 AutoDG Droplet Digital PCR system (Bio-Rad). Primers and probe sequences were as follows: h*NPC1* primer: dCNS361140976, MIQE Context seq1:1-123: +CTCTACAGTTTCTGTCCAGATGTCCATCCTGTTTTTATAACCTACTGAACCTG TTTTGTGAGCTGACATGTAGCCCTCGACAGAGTCAGTTTTTGAATGTTACAGC TACTGAAGATTATGTTG; and *GAPDH* primer: dMmuCNS300520369, MIQE Context mm10|chr6:125161758-125161880:+CCAATAAAGATACAT GCACAAAAGTTGATTGAGCCTGCTTCACCTCCCCATACACACCCTCCCTCCCC CAACACCGCATTAAAACCAAGGAGAGGTGGGTGCAGCGAACTTTATTGA TGGTAT. Each reaction was prepared in a final volume of 20 $\mu$l, consisting of 900 nM forward and reverse primers, 250 nM probe, 1X ddPCR SuperMix for Probes (1863024; Bio-Rad), and template DNA. 0.5–50 ng of DNA was used for gene copy-number quantification with brain (cerebrum, cerebellum, or brainstem) and liver homogenates. Additional organs including spleen, kidney, lung, and leg muscle were assayed for the 10-wk-old cohort in the dose study, using 0.5–5 ng of DNA per reaction. Samples were loaded into DG32 Automated Droplet Generator Cartridges (Bio-Rad). Automated Droplet Generation Oil for Probes (no dUTP) (1864110; Bio-Rad) was then added, and droplets were generated using Automated Droplet Generator (Bio-Rad). Droplets were transferred to a 96-well PCR plate and placed into C1000 Touch Thermal Cycler with 96-Deep Well Reaction Module (Bio-Rad). PCR cycling conditions were as follows: 10 min at 95°C for DNA polymerase activation, followed by 40 cycles of 30 s at 94°C for denaturation and 1 min at 60°C for annealing and extension, and ending at 98°C for 10 min for DNA polymerase deactivation and 4°C for cooling. PCR plates were then loaded into QX200 Droplet Reader (Bio-Rad), and droplet signal was read as being either positive or negative amplification. Data were collected using QuantaSoft software (QuantaSoft v1.7.40917; Bio-Rad) with CNV selection.

## Mass spectrometry imaging and lipidomics

The fresh-frozen half of the brain was sectioned on a CryoStar NX50 Cryostat set to –12°C in preparation for mass spectrometry imaging. The frozen tissue was divided into four to six 10-$\mu$m-thick sections, which were then promptly thaw-mounted onto ITO slides (MIDSCI) and stored at –80°C. Immediately before imaging sections, the slides were removed from the –80°C freezer and washed with ice-cold 50 mM ammonium formate for 20 s, then dried in vacuo. 9-Aminoacridine and 2,5-dihydroxybenzoic acid were chosen as the matrices for negative and positive mode, respectively. One hundred milligrams of solid matrix was dissolved in 10 ml of 50:50 $H_2O$:ACN + 0.2% TFA and filtered using a 0.2-$\mu$m syringe filter. Filtered matrix was applied to the slide using HTX TM-Sprayer.

Mass spectrometry imaging was performed on a Bruker rapifleX MALDI-TOF with a 10-kHz laser set to 60% power, 500 laser shots per pixel, and a step size of 35 $\mu$m. The instrument was operated in negative and positive mode within an m/z range of 200–1,800. All data processing techniques including region-of-interest determination, spatial segmentation, mass spectrum extraction, and image generation were performed using Bruker's SCiLS software. LIPID MAPS and the Human Metabolome Database were used to annotate and identify lipids according to accurate mass measurements. Hedges' g analysis was performed for GM2 lipid species across the experimental groups using total ion count–normalized peak areas.

## Statistical analysis

Randomization was achieved with multiple cohorts. Mice within each cohort were included from each age at the injection group or at each dosage. Statistical analysis was performed using GraphPad Prism version 9.5.1 for Windows or Mac. Normality was evaluated for datasets, and appropriate parametric or nonparametric tests were selected for further analysis. Data are presented as the mean ± SD. The Kaplan–Meier survival curves used the log-rank Mantel–Cox test to assess significance, with a Bonferroni correction applied for $P < 0.0083$ for multiple (six) comparisons. Other statistical tests were as follows: Kruskal–Wallis test with Dunn's multiple comparisons test, one-way ANOVA with Tukey's multiple comparisons test, two-way ANOVA with Tukey's multiple comparisons test, and linear regression test (all multiple comparisons tests use post hoc Bonferroni's correction). In all figures: *$P < 0.05$, **$P < 0.01$, ***$P < 0.001$, ****$P < 0.0001$.

# Data Availability

Mass spectrometry imaging data are available at MassIVE MSV000101131.

# Supplementary Information

# Acknowledgements

We thank Laura L Baxter for her support and instruction in statistical analysis, as well as Stephen Wincovitch for imaging and quantification support. We are grateful for the technical assistance in droplet digital PCR provided by the National Cancer Institute Genomics Core. We sincerely acknowledge the NIH animal care and veterinary staff for the care of mice used in these studies. We also recognize the driving force behind this work: individuals with NPC1 disease. Their perseverance in the face of this debilitating disease inspires and humbles us. This work was supported by the Intramural Research Program of the National Human Genome Research Institute (NHGRI) at the NIH (1ZIAHG000068-16), the *Eunice Kennedy Shriver* National Institute of Child Health and Human Development (NICHD) at the NIH (ZIAHD008988), grants from the NIH (R01NS114413, R01NS124784), and the Ara Parseghian Medical Research Fund at the University of Notre Dame. AV Mylvara, AL Gibson, T Gu, CD Davidson, AA Incao, CP Venditti, and WJ Pavan were supported by the Intramural Research Program at NHGRI; CD Davidson, K Melnyk, SR Gembic, and FD Porter were supported by the Intramural Research Program at NICHD. Additional support for CD Davidson came from the Support of Accelerated Research for NPC (Hide & Seek

Foundation and Dana's Angels Research Trust). D Pierre-Jacques was supported by the Bridge to Doctoral Program and the Diversifying Faculty in Illinois Fellowship.

## Author Contributions

AV Mylvara: conceptualization, data curation, formal analysis, validation, investigation, visualization, and methodology.

AL Gibson: data curation, formal analysis, validation, investigation, visualization, and methodology.

T Gu: data curation, formal analysis, validation, and investigation.

CD Davidson: conceptualization, data curation, formal analysis, validation, investigation, and visualization.

AA Incao: investigation and methodology.

K Melnyk: formal analysis, validation, investigation, and visualization.

SR Gembic: formal analysis, validation, investigation, and writing—review and editing.

D Pierre-Jacques: data curation, formal analysis, investigation, visualization, and methodology.

SM Cologna: conceptualization, resources, and methodology.

CP Venditti: conceptualization and methodology.

FD Porter: resources.

WJ Pavan: conceptualization and resources.

## Conflict of Interest Statement

CP Venditti and WJ Pavan have NIH patents filed on work related to NPC1 genes and the AAV gene therapy treatment of NPC1 (US Patent Publication Numbers 20180104289, 20210113635).

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
