## [Reviewer comments · Life Science Alliance]

Optimization of systemic AAV9 gene therapy in Niemann-Pick disease, type C1 mice

Avani Mylvara, Alana Gibson, Tansy Gu, Cristin Davidson, Art Incao, Katerina Melnyk, Susan Gembic, Dominick Pierre-Jacques, Stephanie Cologna, Charles Venditti, Forbes Porter, and William Pavan

DOI: <https://doi.org/10.26508/lsa.202402874>

Corresponding author(s): Cristin Davidson, Eunice Kennedy Shriver National Institute of Child Health and Human Development and Forbes Porter, NIH

Review Timeline:

Submission Date:	2024-06-06
Editorial Decision:	2024-08-06
Revision Received:	2026-01-28
Editorial Decision:	2026-02-26
Revision Received:	2026-03-14
Accepted:	2026-03-17

Scientific Editor: Tim Fessenden

Transaction Report:

August 6, 2024

Re: Life Science Alliance manuscript #LSA-2024-02874

Dr. Cristin D Davidson
Eunice Kennedy Shriver National Institute of Child Health and Human Development
National Human Genome Research Institute
9000 Rockville Pike
CRC Bldg 10, Rm E1-3288
Bethesda, MD 20892

Dear Dr. Davidson,

Thank you for submitting your manuscript entitled "Optimization of systemic AAV9 gene therapy in Niemann-Pick disease type C1 mice". The manuscript has been evaluated by expert reviewers, whose reports are appended below. Unfortunately, after an assessment of the reviewer feedback, our editorial decision is against publication in Life Science Alliance.

Although your manuscript is intriguing, I feel that the points raised by the reviewers are more substantial than can be addressed in a typical revision period. If you wish to expedite publication of the current data, it may be best to pursue publication at another journal.

Given the interest in the topic, I would be open to re-submission to Life Science Alliance of a significantly revised and extended manuscript that fully addresses the reviewers' concerns and is subject to further peer review. If you would like to resubmit this work to Life Science Alliance, you may submit an appeal directly through our manuscript submission system. Please note that priority and novelty would be reassessed at re-submission.

Regardless of how you choose to proceed, we hope that the comments below will prove constructive as your work progresses.

Thank you for thinking of Life Science Alliance as an appropriate place to publish your work.

Sincerely,

Reviewer #1 (Comments to the Authors (Required)):

This is a study by Mylvara et al further optimizing a systemic AAV9-mediated gene addition therapy for NPC1 disease. While previous studies by this group evaluated varying injection routes at the presymptomatic stage, those findings largely will not translate to the patient NPC population as they are most commonly identified due to symptom onset. Herein, the authors detail first a dose response in the more traditional Npc1m1N model; however, the selection of doses was not done in a clear fold increase fashion (i.e. 5 or 10 fold increase) and the rationale for doses selected is not clear. The authors then select the middle (not most efficacious dose, which again was not clear) and did a time course study in 4 week (pre-symptomatic), 6 week (early-symptomatic), and 8 week (late-symptomatic) Npc1m1N mice in which they demonstrate that earlier intervention is the most efficacious. A downfall to this particular aspect of the study is that the authors selected to evaluate age-matched (9 weeks of age) instead of duration post-treatment matched mice. The authors show that the mice treated at 8 weeks have higher copy number than mice treated at 4 weeks which could be novel information if age does enhance transduction of AAV; however, the mice treated at 4 weeks of age are 5 weeks post treatment, while the mice treated at 8 weeks are only 1 week post treatment. In order to give the vector equal amount of time to express in each cohort, they should have been treatment duration matched, not age-matched (ex all mice taken down 4 weeks post treatment). The authors then evaluate the medium dose treatment in a new, more patient orthologous mouse model, Npc111061T. Overall, the establishment of a dose and time window is particularly important for diseases that are not on the RUSP, as the vast majority of children are diagnosed symptomatically. However, the study has experimental design elements including dose selection and age-matching that lessen enthusiasm for this study. The paper is in general well written and figures are clear; however, accuracy between text and figures needs to be improved. Major and minor comments are detailed below:

- In the results, please clarify which mouse model is being treated. At the end of the introduction the authors introduce a second

model, Npc111061T, and then the results start describing gene therapy treated mice; however, this first set of results is the m1N model but that is not detailed in the text.

- It is not clear how the dose escalation was selected (i.e. why not a 5 or 10 fold dose increase?).
- The AAV9-treated mice have a significantly extended lifespan, up to ~35 weeks with the high dose. Can the authors detail what the mice are dying from as this is still a premature lifespan compared to normal mice. This cause, limitation, and potential improvements should be included in the discussion.
- There is substantial variability in the weight of the medium and high dose treated mice (but not the untreated and low dose treated mice). Can the authors comment on this. Is there evidence of variability in efficiency of the injection route? This would again be another good point for the discussion section.
- This statement could be further clarified or separated in to two analyses, significance from vehicle then significance from normal. High dose mice being "like" normal mice is not specific. Were only the high dose mice not significantly different from normal mice? Line 167: "Between 6 and 9 weeks, all treatment groups were significantly delayed in phenotype onset from vehicle treated, while high dose treated mice were like Npc1+/-."
- If Purkinje cells are primary target and analyzed in Figure 3, then why was cerebrum and not cerebellum chosen for analysis of vector copy number in Figure 2 data set?
- The relevance of Figure 2 C is not clear as the manuscript is focused on defining a dose response.
- Figure 2Aiii and 2Biii are not discussed in the text.
- The axis on 2Aiii should be changed so that the differences between groups can be visualized. The axis should not be more than double the highest data point.
- For Figure 2D it would be helpful to have the cerebrum and liver in comparison to the other brain regions and organs. Consider combining 2Ai, 2Bi, and 2D to one figure (will likely need an axis break for the liver).
- This is a confusing figure call out because Fig 2C is VCN not protein Line 200 "though cerebrum from high dose treated mice revealed very low, albeit detectable levels of protein (Fig 2C)."
- Figure 2F is cited 2 times (Lines 196-203) but there is no Figure 2F.
- Figure 3B the text states that staining was done with CD68 (211), the figure shows Iba1.
- It is not clear why 3B is a separate analysis from 3A and different brain sections were used.
- It is not clear why 3F and G copy number data is with histology and not figure 2 VCN data.
- Is 4B also CD68 staining? It is not labeled. If so what is the difference in 4A middle row and 4B? Which images is 4C quantifying? To the eye, in 4B the high dose does not appear to have less staining (CD68?) than the mid-dose. Perhaps a representative area is not being shown?
- In general, the authors need to review figures and the corresponding text and improve accuracy, labeling, and appropriate text call outs to figures.
- Lines 243-247 reintroduce the Npc111061T model, but then data in the subsequent paragraph is still m1N mice. In general the authors need more clarity on which model is being shown.
- Figure S8 should be reconsidered as a main figure instead of supplementary. Is there any quantification associated with the mass spec imaging, if so it should be included?
- It is not clear why the medium and not the high dose was selected for the time course study detailed in Figure 5.
- In the time course study, it would be helpful if the authors detailed the phenotypic differences between early and late symptomatic.
- For the time course study, it is not clear why the mice are age-matched instead of time post-treatment matched. The authors are showing that the mice treated at 8 weeks have higher copy number than mice treated at 4 weeks; however, the mice treated at 4 weeks are now 5 weeks post treatment, the mice treated at 6 weeks are 3 weeks post treatment, and the mice treated at 8 weeks are only 1 week post treatment. In order to give the vector equal amount of time to express in each cohort, the mice

should have been treatment duration matched, not age-matched (ex all mice taken down 4 weeks post treatment).

- It is not clear why the medium and not the high dose was selected for the Npc111061T model detailed in Figure 7.
- In the discussion, this statement is out of date. The SMA gene therapy was FDA approved in 2019 and clinical trial results are completed: Line 411, several clinical trials using peripherally administered gene therapy for neurological diseases are well underway (SMA1 NCT03461289,"
- The discussion largely restates the results instead of further discussing particular aspects of the study (strengths, weaknesses, limitations, unexpected findings) in greater detail/speculation.

Reviewer #2 (Comments to the Authors (Required)):

Authors reported about the results of gene therapy for NPC1 deficient mice by systemic injection of AAV9-EF1a(s)-hNPC1, changing doses and treatment day. They used three doses from low to high, 7.87×10^{12} vector genomes/kg, 1.28×10^{14} vg/kg, or 3.06×10^{14} vg/kg, and showed highest dose was most effective. And they treated mice on 4-, 6-, and 8-week-old, and showed early treatment was better.

The examination was well organized and well analyzed.

However, the objective and plans of examination were not significant, no meaning.

They used highest dose as 3.06×10^{14} vg/kg. In the clinical trial of gene therapy for X-linked myotubular myopathy using AAV8 vector, three patients died at the dose of 3×10^{14} vg/kg. It has been considered that high dose intravenous injection of AAV has risk for liver dysfunction, thrombocytopenia, thrombotic microangiopathy and death, and about 1×10^{14} vg/kg, the dose of Zolgensma may be limit. If the authors want to use high dose iv, nonhuman primate should be used considering immune suppression, not by mice. Even in high dose, the survival and results were almost same with previous reports.

Authors also tried the treatment at later stage. NPC1 is a neurodegenerative disease. Neurons have gradually disappeared, and treatments after neuronal loss show no effect. It is obvious without examination. Earlier treatment showed better effects even in AADC deficiency that is not degenerative disorder. We have to develop early diagnosis methods to treat patients earlier.

Reviewer #3 (Comments to the Authors (Required)):

The manuscript addresses a fundamental challenge for biomedical research: to develop efficient therapies for the numerous inherited and ultimately fatal lysosomal disorders presenting progressive neurologic symptoms and lacking curative treatment. This specific study addresses Niemann-Pick type C disease. Most cases are provoked by variants of NPC1, a membrane-resident cholesterol transporter acting in concert with - largely unknown - modifiers that determine the highly variable disease onset, symptoms and life span. The study focuses on gene therapy based on viral vectors. Within the last decades, this approach has gained new momentum for many diseases due to substantial advances. The present study follows up on groundbreaking work by the authors and other groups showing that AAV-based gene therapy can halt disease progression in an established mouse model deficient of NPC1. Here, the authors have performed an impressive and comprehensive range of experiments that address key questions in a largely satisfactory manner. The new results represent another milestone as they reveal dose- and age-dependent effects of treatment and thus allow to optimize treatment paradigms. Moreover, they show efficacy in a second mouse model showing a more prolonged disease progression. These findings are very important for the field and beyond, notably for lysosomal diseases that are caused by variants of membrane-resident proteins. Overall, the dataset is nearly complete and the manuscript is of outstanding interest. Unfortunately, however, the presentation of data precludes immediate publication. The description of results, the discussion and the organisation of figures show severe weaknesses that diminish appreciation and understanding of the data. The authors are strongly advised to critically re-read their text and look again at their figures, and to revise both. Specifically, they may consider the following points to improve the quality and readability of their manuscript:

- There is a lack of information about the sex of animals except for weight measures. The authors should indicate whether there are sex-specific differences in the different parameters analysed. Moreover, the authors could indicate in relevant graphs the sex of individual animals by different symbols.

- Lines 49-51: The authors should rewrite this sentence. Essentially, their study addresses the dose- and age-dependent effects of gene therapy. The term "window of therapeutic efficacy" anticipates the results. Moreover, the term "early symptomatic" appears as odd and "post-symptomatic" is potentially misleading. A suggestion could be "before onset of neurologic symptoms and at an early and late stage of disease progression".

- Line 93: The phrase "in a severe model" should be modified, and the mouse line should be described for non-expert readers.

Evidently, the model is not severe, but its phenotype.

- Lines 113-117. The two-part sentence appears a bit long, the authors may consider to split starting the second sentence with "However..."

- Lines 124-125: The authors should consider to move the sentence to the last paragraph of the Introduction.

- Line 139: For clarity, change to "a single retro-orbital".

- Lines 139-368: The authors should avoid lab jargon and streamline the text to enhance comprehension. For example: lines 161-174, the description of the Results appears as verbose, and unstructured. In lines 163-165, the authors describe "significant" changes, whereas the statistical test is described only in the following sentence. Moreover, the effects of the different doses are described in the following statement. The separate description of effects at different ages (lines 169-174) appears as repetitive. Together, this is somewhat tiring, and diminishes the understanding and appreciation of the findings. Moreover, terms describing methods should be used in a consistent manner, for example "immunohistochemistry" or "immunohistochemical staining" rather than imprecise terms/lab jargon like "immunofluorescence" or "immunostaining" etc.

- Notwithstanding personal taste and style, the text contains many imprecise or odd terms and statements that should be replaced. The authors should search through their text and revise accordingly. Here are some examples:

- - Lines 166-167: the term "the vehicle trajectory".

- - Line 177 "greater viral transduction".

- - Line 183 and others: "predictive relationship" or "significant relation". The authors used selected tests to determine whether there is a correlation between parameters. This should be described correctly.

- - Lines 193-194: "an increased dose (particularly high dose)..."

- - Line 209: "Immunofluorescent staining of the brain and immunohistochemical staining of the liver" is imprecise and needs to be corrected to describe exactly the experiments that were performed.

- - Line 249: "gene therapy administration"

- - Line 315: "loss of Purkinje neuron survival"

- - Line 377: "define the durability"

- - Line 391: "many NPC1 individuals are severe"

- - Line 620: "varies depends on"

- Lines 149-159 & 279-284 & 340-343, Figs. 1, S1A,B, 5, 7: The parameter "week of peak weight" and its physiologic/clinical relevance is unclear. Does this add any information compared to the "percent weight change", which is clearly more accessible and significant? What if a specific animal does not present a week of peak weight following treatment onset? The authors should consider to remove the "week of peak weight". On the other hand, the development of body weight following the different treatments is a very important piece of information. Therefore, the corresponding graphs (now in Fig. S1A-D) should be shown in the corresponding main figures. As suggested above, the authors should tidy the description in order to transmit the message more clearly. The reference to figure S2C,D is wrong, it should be corrected to S1.

- Lines 177-203 & 303-: The parallel measurements of protein and DNA in the same sample is a great plus, but the authors do not expose this feat sufficiently. As is, the description of immunoblot results reads like a last minute add-on rather than an integral part of the analysis. The authors should modify this and introduce these data more appropriately.

- Line 187: The abbreviation CNV should be defined here.

- Line 206: The term "brain" should be replaced by "cerebellum", since no other brain area is described.

- Lines 209-214: Another example of a somehow confusing description: the authors describe their immunohistochemical markers, switch to filipin histochemistry and then go back to immunohistochemical and immunoblots. This should be modified.

- Lines 209-246, Figs. 3A, 4A: The histochemical staining with filipin is an important element, but without a quantitative assessment of fluorescence intensities across different animals, the robustness and reproducibility of the data remains unclear.

- Lines 211,212 & 233,234: The authors should consider to describe the marker CD68 in a more concise manner. Their statements that CD68 labels microglia and myeloid cells together with the sentence "..., many of which are macrophages like Kupffer cells..." can confuse the non-expert reader.

- Lines 214-216: Despite repeated statements in the literature, GFAP cannot be considered a marker of neuroinflammation (see Escartin et al., 2021 Nature Neurosci). An increase of the protein suggests the presence of reactive astrocytes. Notably, in the cerebellum, most of the GFAP-positive cells are probably reactive Bergmann glia as indicated by their location and shape (Fig. 3, higher mag images). Corresponding statements should be corrected.

- Lines 226-231 and Fig. 3F, G: These results do not seem to fit in here. They should be described in the paragraph lines 177-

203 and figure 2, where similar results are presented for cerebrum and liver.

- Lines 243-247: The meaning of this paragraph is unclear. Up to this point, only dose dependence was described, but no data on age or the hypomorphic model.
- Line 249-253: The title and introductory statements should mention right away in which organ or tissue the lipid distribution was analysed.
- Lines 249-265: Without proper quantification, the relevance of the data is unclear. As is, the dataset appears as dispensable. How many animals and sections per animal were analysed? The figure number S8 is not in line with the figures mentioned in the previous chapter.
- Lines 260-264: The meaning of these statements is unclear. They should be rewritten.
- Line 268: The title appears long and verbose. The authors should consider to modify. The term describing the viral construct is used in an inconsistent manner in the different chapter titles.
- Line 319: Is there a specific reason why the markers used to detect dose effects in the cerebellum were not used to reveal age-dependent differences in this brain area?
- Line 322: Replace "In the liver, higher copy numbers predicted" by "Higher copy numbers in liver cells predicted".
- Lines 303-327: The description of the results is not well structured. As outlined for other paragraphs, the authors should tidy the text.
- Lines 310-312 & 323-325, Fig. 6, S3A: The copy numbers seem to be higher the shorter the interval between vector administration and measurement. Does this mean that copy numbers decrease with time following injection?
- Line 349: The author mention "untreated mice". Do their cohorts include any untreated = non-injected animals? This should be clarified.
- Line 353: "though were still": a subject is missing.
- Lines 363-368: The staining of myeloid cells can not indicate "liver lipid storage burden" or cholesterol accumulation. This should be corrected. The statements obviously raise the question whether the authors performed filipin staining in these animals.
- Line 393: Define "RUSP"!
- Line 371-422: Similar as for the Results, the authors should consider to revise the Discussion. As is, it appears as a bit patchy with unnecessary repetition. Moreover, some topics are not really discussed in the light of previous studies. Just as examples: lines 390-391 stand isolated, repeating what has been written in lines 380-381 and lines 394-395 mention the goal that has already been presented before. The discussion of gene copy number and outcome is superficial. Can this be compared to previous preclinical studies on NPC or other diseases? What do the present results mean for preclinical R&D in other diseases with defects in lysosomal membrane-spanning proteins?
- Lines 488, 493 & 552: The authors mention "spleen, kidney, lung, and leg muscle" as well as "brainstem", but no data are shown for these organs and brain region.
- Lines 607-685: The authors should revise the figure legends. A few examples:
 - - Line 634: The title appears as imprecise not referring to the actually organ or tissue.
 - - Line 636: Change to "astrocytes and Bergmann glia".
 - - Lines 638: The IBA1 staining is mentioned in the legend, but not in the corresponding chapter of the Results. Can IBA1 really be considered a surrogate marker for neuroinflammation? Again, the authors should introduce the marker properly.
- Figures: The authors should homogenize the design and size of graphs and to arrange panels reducing white space.
- Fig. 1: The authors should consider to remove the table from the figure. The statistics can easily be integrated in panel A by asterisks. The number of animals is already evident from panel A.
- Fig. 2: The rectangles surrounding panels should be removed. Moreover, the y axis scaling in panel Aiii should be modified (max at 0.1) to show the differences between treated groups. In addition, the authors should consider to move the representative Western blots from Fig. S5 to this main figure.

- Fig. 2C: It is unclear what the lines represent? Linear regression with 2 data points seems to make little sense. Where are green data points representing high dose? What is the difference to the graph shown in panel 2Aii?
- Fig. S2: The figure is not a figure, it's a table.
- Fig. S3. The data seem important, therefore they should be integrated in the main figure.
- Fig. S5: The figure contains legend text that should be removed.

January 23, 2026

Dear Dr. Sawey, PhD,

The authors of manuscript #LSA-2024-02874 have requested an appeal. Their comments are below.

Good day,

We kindly request approval for resubmission for our body of work highlighting optimization of AAV9 gene therapy as a treatment for Niemann-Pick type C1 disease. Included are several relevant files and should LSA be open to a resubmission, we will upload all documents as per the submission process. Many thanks for considering our request! We look forward to sharing these important findings with the scientific community through a well-respected, peer-reviewed journal such as LSA.

Best, Cristin Davidson

You can accept or decline this request from the manuscript using the following link:

<https://lsa.msubmit.net/cgi-bin/main.plex?el=A6Na2BQe6A5UhX3F3A9ftd5cXSQXnC76kSAgeFGq0kgZ>

Sincerely,

Editorial Staff

January 26, 2026

MS: LSA-2024-02874

Dr. Cristin D Davidson
Eunice Kennedy Shriver National Institute of Child Health and Human Development
National Human Genome Research Institute
9000 Rockville Pike
CRC Bldg 10, Rm E1-3288
Bethesda, MD 20892

Dear Dr. Davidson,

Thank you for your correspondence including the revised manuscript entitled, "Optimization of systemic AAV9 gene therapy in Niemann-Pick disease type C1 mice" as well as your rebuttal letter. Our prior decision has been reconsidered and I am pleased to let you know that we have decided to invite the revised manuscript for re-review. Please note we will be looking for strong reviewer support of the revised work in order to proceed towards publication.

Having approved your appeal, you will now be able to upload the revised manuscript as a resubmission using the link below.

****Please note your manuscript will not be sent for re-review until you upload it to our submission system.****

Please use the following link to submit your revised manuscript:

<https://lsa.msubmit.net/cgi-bin/main.plex?el=A7Na4BQe2A4CkSr2l1B9ftdFPixY7QofNQqtO3mC3E4QZ>

Yours sincerely,

AV Mylvara et al, resubmission
Optimization of systemic AAV9 gene therapy in Niemann-Pick disease, type C1 mice
January 23, 2026

Reviewers' Comments & Responses

Please find comments from each reviewer along with our responses in blue text following each comment/suggestion. Given the extensive editing and revision, we did not attach a track changes document of the original but rather reference the appropriate line numbers in the updated manuscript for each response.

Reviewer #1

This is a study by Mylvara et al further optimizing a systemic AAV9-mediated gene addition therapy for NPC1 disease. While previous studies by this group evaluated varying injection routes at the presymptomatic stage, those findings largely will not translate to the patient NPC population as they are most commonly identified due to symptom onset. Herein, the authors detail first a dose response in the more traditional Npc1m1N model; however, the selection of doses was not done in a clear fold increase fashion (i.e. 5 or 10 fold increase) and the rationale for doses selected is not clear. The authors then select the middle (not most efficacious dose, which again was not clear) and did a time course study in 4 week (pre-symptomatic), 6 week (early-symptomatic), and 8 week (late-symptomatic) Npc1m1N mice in which they demonstrate that earlier intervention is the most efficacious. A downfall to this particular aspect of the study is that the authors selected to evaluate age-matched (9 weeks of age) instead of duration post-treatment matched mice. The authors show that the mice treated at 8 weeks have higher copy number than mice treated at 4 weeks which could be novel information if age does enhance transduction of AAV; however, the mice treated at 4 weeks of age are 5 weeks post treatment, while the mice treated at 8 weeks are only 1 week post treatment. In order to give the vector equal amount of time to express in each cohort, they should have been treatment duration matched, not age-matched (ex all mice taken down 4 weeks post treatment). The authors then evaluate the medium dose treatment in a new, more patient orthologous mouse model, Npc111061T. Overall, the establishment of a dose and time window is particularly important for diseases that are not on the RUSP, as the vast majority of children are diagnosed symptomatically. However, the study has experimental design elements including dose selection and age-matching that lessen enthusiasm for this study. The paper is in general well written and figures are clear; however, accuracy between text and figures needs to be improved. Major and minor comments are detailed below:

- In the results, please clarify which mouse model is being treated. At the end of the introduction the authors introduce a second model, Npc111061T, and then the results start describing gene therapy treated mice; however, this first set of results is the m1N model but that is not detailed in the text.

Each results section title now specifies which mouse model is being described.

- It is not clear how the dose escalation was selected (i.e. why not a 5 or 10 fold dose increase?). The dose selection rationale is now described in detail in the methods (line 575-585) and at the beginning of the Results section in lines 134-141. In short, doses were selected based on ACUC approved volumes and concentration of vector preparations plus doses being used in humans at the time study was being conducted.

- The AAV9-treated mice have a significantly extended lifespan, up to ~35 weeks with the high dose. Can the authors detail what the mice are dying from as this is still a premature lifespan compared to normal mice. This cause, limitation, and potential improvements should be included in the discussion.

The disease does eventually progress, and mice die from these complications – gene therapy does not halt disease progression. This is included in the discussion in lines 419-420 and 457-463.

- There is substantial variability in the weight of the medium and high dose treated mice (but not the untreated and low dose treated mice). Can the authors comment on this. Is there evidence of variability in efficiency of the injection route? This would again be another good point for the discussion section.

Potential reasons for variability in the readout metrics, specifically weight, have been added to the discussion (lines 468-480).

- This statement could be further clarified or separated into two analyses, significance from vehicle then significance from normal. High dose mice being "like" normal mice is not specific. Were only the high dose mice not significantly different from normal mice? Line 167: "Between 6 and 9 weeks, all treatment groups were significantly delayed in phenotype onset from vehicle treated, while high dose treated mice were like *Npc1*^{+/+}."

This language has been clarified in the results section titled "*Npc1*^{m^{1N}} mice treated with higher doses of AAV9-EF1a(s)-hNPC1 gene therapy show increased survival and delayed disease progression" in lines 171-176. Language now refers to lower and higher composite phenotype scores.

- If Purkinje cells are primary target and analyzed in Figure 3, then why was cerebrum and not cerebellum chosen for analysis of vector copy number in Figure 2 data set?

Figure 2 and 3 have been restructured and cerebellum is now included in Figure 2.

- The relevance of Figure 2 C is not clear as the manuscript is focused on defining a dose response.

This figure panel has been removed.

- Figure 2Aiii and 2Biii are not discussed in the text.

Figures have been restructured and the linear regressions are now included in the results section (lines 213-216).

- The axis on 2Aiii should be changed so that the differences between groups can be visualized. The axis should not be more than double the highest data point.

This figure is now 2Bii and the axis has been changed.

- For Figure 2D it would be helpful to have the cerebrum and liver in comparison to the other brain regions and organs. Consider combining 2Ai, 2Bi, and 2D to one figure (will likely need an axis break for the liver).

A comparison of *hNPC1* copy number from all organs is now included in supplemental figure 2.

- This is a confusing figure call out because Fig 2C is VCN not protein Line 200 "though cerebrum from high dose treated mice revealed very low, albeit detectable levels of protein (Fig 2C)."

The Fig 2C panel has been changed, where copy number is shown in column (i) and protein level in column (ii) of Figure 2.

- Figure 2F is cited 2 times (Lines 196-203) but there is no Figure 2F.

Figure 2 was restructured and all graphs depicted in Figure 2 are contained in the results text (lines 198-202 for initial sub-figure callouts).

- Figure 3B the text states that staining was done with CD68 (211), the figure shows Iba1.

In lines 241-244 we now describe the use of IBA1 in staining and CD68 antibody for Western blots in Fig 3B and 3E, respectively.

- It is not clear why 3B is a separate analysis from 3A and different brain sections were used.

Technical limitations of combining multiple antibodies and filipin labeling necessitate two separate analyses for the IBA1 and cholesterol/GFAP/calbindin D. Also, filipin cannot be used on formalin fixed paraffin embedded (FFPE) sections while several of the antibodies used in this work are. Therefore, we used both formalin fixed vibratome-cut free floating sections and FFPE sections mounted to slides for immunofluorescence work. Type of cut tissue is correctly described in the figure legend (lines 738-741).

- It is not clear why 3F and G copy number data is with histology and not figure 2 VCN data.

This has been restructured so that this 3F, G are now part of Figure 2.

- Is 4B also CD68 staining? It is not labeled. If so what is the difference in 4A middle row and 4B?

Which images is 4C quantifying? To the eye, in 4B the high dose does not appear to have less staining (CD68?) than the mid-dose. Perhaps a representative area is not being shown?

These are both CD68 staining and are now labeled, the only difference between the two is the type of staining used for CD68 labeling – immunohistochemical (which was used for quantification) and immunofluorescence (used for colabeling with filipin for unesterified cholesterol). These are representative images, which is now specified in the figure legend in lines 748-752.

- In general, the authors need to review figures and the corresponding text and improve accuracy, labeling, and appropriate text call outs to figures.

We appreciate the feedback and we believe all discrepancies or ambiguous text has now been clarified.

- Lines 243-247 reintroduce the *Npc111061T* model, but then data in the subsequent paragraph is still *m1N* mice. In general the authors need more clarity on which model is being shown.

Each results section title now specifies which mouse model being discussed.

- Figure S8 should be reconsidered as a main figure instead of supplementary. Is there any quantification associated with the mass spec imaging, if so it should be included?

We have respectfully chosen to keep this figure as supplemental data. While the images show a trend towards higher doses bringing lipid levels closer to *Npc1*^{+/+} levels and GM2 ganglioside was quantified, there was no significant difference for the GM2 levels visualized by mass spectrometry. We consider this better left as supporting data given that many other metrics presented in the main figures show statistically significant changes.

- It is not clear why the medium and not the high dose was selected for the time course study detailed in Figure 5.

We selected the medium dose for comparisons because it aligns with our previously published studies, ensuring consistency and comparability (Chandler et al 2017, Davidson et al 2021). Using a medium dose allows for more dynamic analysis of dose-response effects, avoiding saturation at the upper limit. Lastly, due to material constraints, we did not have enough of the more concentrated vector preparation used for the high dose to include it in all comparisons. Dose selection is now described at in the methods (line 574-585) and at the beginning of the Results section (lines 132-141).

- In the time course study, it would be helpful if the authors detailed the phenotypic differences between early and late symptomatic.

Symptom onset is now described in results section titled “AAV9 treatment at 4-weeks improves survival and delays disease progression in *Npc1*^{m1N} compared to treatment at 6- or 8- weeks of age” in lines 280-282.

- For the time course study, it is not clear why the mice are age-matched instead of time post-treatment matched. The authors are showing that the mice treated at 8 weeks have higher copy number than mice treated at 4 weeks; however, the mice treated at 4 weeks are now 5 weeks post treatment, the mice treated at 6 weeks are 3 weeks post treatment, and the mice treated at 8 weeks are only 1 week post treatment. In order to give the vector equal amount of time to express in each cohort, the mice should have been treatment duration matched, not age-matched (ex all mice taken down 4 weeks post treatment).

Our primary focus was to detect survival differences, then to examine pathological manifestations rather than solely the effect of copy number, which precluded the use of a single post-dose interval based timepoint for all experiments. We recognize it as a limitation of the study, and have included this limitation in the discussion and suggest that future studies might explore interval-based analyses to better account for dynamic vector expression changes over time (lines 482-489).

- It is not clear why the medium and not the high dose was selected for the *Npc1*11061T model detailed in Figure 7.

We selected the medium dose for comparisons because it aligns with previously published studies, ensuring consistency and comparability. Due to material constraints, we did not have enough of the high dose to include it in all comparisons (see earlier response to dose selection, 3 bullet points above).

- In the discussion, this statement is out of date. The SMA gene therapy was FDA approved in 2019

and clinical trial results are completed: Line 411, several clinical trials using peripherally administered gene therapy for neurological diseases are well underway (SMA1 NCT03461289," We appreciate this correction and text has been modified in the Introduction (lines 82-85).

- The discussion largely restates the results instead of further discussing particular aspects of the study (strengths, weaknesses, limitations, unexpected findings) in greater detail/speculation. Again, we welcome the constructive feedback and have restructured the discussion to include these points.

Reviewer #2:

Authors reported about the results of gene therapy for NPC1 deficient mice by systemic injection of AAV9-EF1a(s)-hNPC1, changing doses and treatment day.

They used three doses from low to high, 7.87×10^{12} vector genomes/kg, 1.28×10^{14} vg/kg, or 3.06×10^{14} vg/kg, and showed highest dose was most effective. And they treated mice on 4-, 6-, and 8-week-old, and showed early treatment was better.

The examination was well organized and well analyzed.

However, the objective and plans of examination were not significant, no meaning.

They used highest dose as 3.06×10^{14} vg/kg. In the clinical trial of gene therapy for X-linked myotubular myopathy using AAV8 vector, three patients died at the dose of 3×10^{14} vg/kg. It has been considered that high dose intravenous injection of AAV has risk for liver dysfunction, thrombocytopenia, thrombotic microangiopathy and death, and about 1×10^{14} vg/kg, the dose of Zolgensma may be limit. If the authors want to use high dose iv, nonhuman primate should be used considering immune suppression, not by mice. Even in high dose, the survival and results were almost same with previous reports.

Authors also tried the treatment at later stage. NPC1 is a neurodegenerative disease. Neurons have gradually disappeared, and treatments after neuronal loss show no effect. It is obvious without examination. Earlier treatment showed better effects even in AADC deficiency that is not degenerative disorder. We have to develop early diagnosis methods to treat patients earlier. With regards to the reviewer's initial point about dosing, this is now included in the discussion in lines 448-455. Our high dose is higher than Zolgensma, but as the reviewer states, this dose in mice is not directly translatable to humans. The value of dose escalation studies is included in the discussion and is compared to the myotubular myopathy study and Zolgensma. NPC1 gene therapy studies in other groups have not assessed treatment at a juvenile age when pathological changes are present (4 weeks of age) or later in life when early or pronounced phenotypic manifestations of disease are present (6 and 8 weeks of age, respectively), so addressing the effect of dose at these stages of disease progression is unique.

With regards to the point made about early diagnosis methods, we are in absolute agreement that early diagnosis is critical to treating patients earlier. However, it remains that NPC1 individuals are typically diagnosed after symptom onset and with years of diagnostic delay, so understanding the therapeutic efficacy of gene therapy at various stages of symptom progression is relevant to the current clinical landscape. This is included in the discussion in lines 430-434.

Finally, we respectfully disagree that this work has minimal significance and meaning. While the results largely support commonsense hypotheses, these studies need to be carried out to confirm the expected outcomes and importantly, to provide convincing, reproducible data to regulatory agencies as we move gene therapy for NPC1 towards clinical trial. In discussions with consultants who regularly interact with the FDA on gene therapy investigational new drug applications, they have made it clear that although the NPC community has access to a large animal model (NPC1 cats), the exact product to be used in humans must be evaluated in a model system, preferably *in*

vivo if an amenable model exists. Feline models of disease are limited in gene therapy studies because use of the human gene is toxic to the cats and work must be carried out with vectors containing the feline gene of interest. Thus, we are left with murine models of NPC1 disease, all of which replicate numerous aspects of NPC disease in humans ranging from pathology to phenotypic manifestations to pathological genetic variants.

Reviewer #3

The manuscript addresses a fundamental challenge for biomedical research: to develop efficient therapies for the numerous inherited and ultimately fatal lysosomal disorders presenting progressive neurologic symptoms and lacking curative treatment. This specific study addresses Niemann-Pick type C disease. Most cases are provoked by variants of NPC1, a membrane-resident cholesterol transporter acting in concert with - largely unknown - modifiers that determine the highly variable disease onset, symptoms and life span. The study focuses on gene therapy based on viral vectors. Within the last decades, this approach has gained new momentum for many diseases due to substantial advances. The present study follows up on groundbreaking work by the authors and other groups showing that AAV-based gene therapy can halt disease progression in an established mouse model deficient of NPC1. Here, the authors have performed an impressive and comprehensive range of experiments that address key questions in a largely satisfactory manner. The new results represent another milestone as they reveal dose- and age-dependent effects of treatment and thus allow to optimize treatment paradigms. Moreover, they show efficacy in a second mouse model showing a more prolonged disease progression. These findings are very important for the field and beyond, notably for lysosomal diseases that are caused by variants of membrane-resident proteins. Overall, the dataset is nearly complete and the manuscript is of outstanding interest. Unfortunately, however, the presentation of data precludes immediate publication. The description of results, the discussion and the organisation of figures show severe weaknesses that diminish appreciation and understanding of the data. The authors are strongly advised to critically re-read their text and look again at their figures, and to revise both. Specifically, they may consider the following points to improve the quality and readability of their manuscript:

- There is a lack of information about the sex of animals except for weight measures. The authors should indicate whether there are sex-specific differences in the different parameters analysed. Moreover, the authors could indicate in relevant graphs the sex of individual animals by different symbols.

Across parameters we analyzed, we did not find sex specific differences. This has been addressed in the initial results section (lines 148-151). We have also included the sample size per sex for weight curves in Table S4A,B,C. Since all mice were included in the weight curves (age-matched cohorts at 9, 10, and 14 weeks plus the survival cohort), this data denotes the total samples size used in these studies.

- Lines 49-51: The authors should rewrite this sentence. Essentially, their study addresses the dose- and age-dependent effects of gene therapy. The term "window of therapeutic efficacy" anticipates the results. Moreover, the term "early symptomatic" appears as odd and "post-symptomatic" is potentially misleading. A suggestion could be "before onset of neurologic symptoms and at an early and late stage of disease progression".

This has been rephrased to early-stage and late-stage disease progression in lines 51-52. We have also defined the term "pre-symptomatic" as before onset of neurologic symptoms as suggested (lines 278-279).

- Line 93: The phrase "in a severe model" should be modified, and the mouse line should be described for non-expert readers. Evidently, the model is not severe, but its phenotype. This has been amended to state the protein truncation results in a severe disease phenotype (lines 93-95).

- Lines 113-117. The two-part sentence appears a bit long, the authors may consider to split starting the second sentence with "However..."
Addressed in line 110 by editing as reviewer suggests.

- Lines 124-125: The authors should consider to move the sentence to the last paragraph of the Introduction.
Addressed in lines 124-128.

- Line 139: For clarity, change to "a single retro-orbital".
Addressed in line 147.

- Lines 139-368: The authors should avoid lab jargon and streamline the text to enhance comprehension. For example: lines 161-174, the description of the Results appears as verbose, and unstructured. In lines 163-165, the authors describe "significant" changes, whereas the statistical test is described only in the following sentence. Moreover, the effects of the different doses are described in the following statement. The separate description of effects at different ages (lines 169-174) appears as repetitive. Together, this is somewhat tiring, and diminishes the understanding and appreciation of the findings. Moreover, terms describing methods should be used in a consistent manner, for example "immunohistochemistry" or "immunohistochemical staining" rather than imprecise terms/lab jargon like "immunofluorescence" or "immunostaining" etc.
Language in this section has been streamlined and better structured in the referenced subsection (lines 144-193). Qualitative descriptions are made before quantitative descriptions where statistical tests are mentioned.
Immunofluorescence staining and immunohistochemical staining are distinct procedures with different applications in our work, so that language has been kept in the results section in lines 247-248 and figure legends. However, we have removed "immunostaining".

- Notwithstanding personal taste and style, the text contains many imprecise or odd terms and statements that should be replaced. The authors should search through their text and revise accordingly. Here are some examples:
 - - Lines 166-167: the term "the vehicle trajectory".
 - - Line 177 "greater viral transduction".
 - - Line 183 and others: "predictive relationship" or "significant relation". The authors used selected tests to determine whether there is a correlation between parameters. This should be described correctly.
 - - Lines 193-194: "an increased dose (particularly high dose)..."
 - - Line 209: "Immunofluorescent staining of the brain and immunohistochemical staining of the liver" is imprecise and needs to be corrected to describe exactly the experiments that were

performed.

- - Line 249: "gene therapy administration"
- - Line 315: "loss of Purkinje neuron survival"
- - Line 377: "define the durability"
- - Line 391: "many NPC1 individuals are severe"
- - Line 620: "varies depends on"

The language called out by the reviewer has been corrected to be clear and accurate. We have carefully reviewed and edited the text with the reviewer's comments in mind.

- Lines 149-159 & 279-284 & 340-343, Figs. 1, S1A,B, 5, 7: The parameter "week of peak weight" and its physiologic/clinical relevance is unclear. Does this add any information compared to the "percent weight change", which is clearly more accessible and significant? What if a specific animal does not present a week of peak weight following treatment onset? The authors should consider to remove the "week of peak weight". On the other hand, the development of body weight following the different treatments is a very important piece of information. Therefore, the corresponding graphs (now in Fig. S1A-D) should be shown in the corresponding main figures. As suggested above, the authors should tidy the description in order to transmit the message more clearly. The reference to figure S2C,D is wrong, it should be corrected to S1.

Week of peak weight is a way to evaluate disease onset, while percent weight change, describes disease progression. This rationale is now described in the results section in lines 178-180.

We feel that Supplemental Fig 1A-F should remain as supplemental because it is difficult to perform statistics and draw conclusions from data with increasingly smaller sample sizes resulting from mice reaching humane endpoint at various ages. The varied ages when *Npc1^{m1N}* mice reach humane endpoint in the survival cohorts generates unexpected increases in the weight curve at older ages. Moreover, we have a large amount of data and have opted to present the strongest data in the main figures of the manuscript while still providing the full breadth of outcome metrics by including supplemental figures.

Figure references have been corrected.

- Lines 177-203 & 303-: The parallel measurements of protein and DNA in the same sample is a great plus, but the authors do not expose this feat sufficiently. As is, the description of immunoblot results reads like a last minute add-on rather than an integral part of the analysis. The authors should modify this and introduce these data more appropriately.

This is more directly addressed as parallel assessments in the results section in lines 200-202, 225, 324-325, and 385-387. Figures 2, 6, and 9 were also adjusted to show parallel assessments of DNA and protein. We appreciate the very helpful suggestion.

- Line 187: The abbreviation CNV should be defined here.

Use of CNV has been replaced with just copy number throughout.

- Line 206: The term "brain" should be replaced by "cerebellum", since no other brain area is described.

This results section has been restructured, where Figure 2 refers to DNA/protein for cerebellum, cerebrum, and liver, while Figure 3 looks at cerebellar pathology. Results sections titles now

reflect these changes in lines 196 and 220.

- Lines 209-214: Another example of a somehow confusing description: the authors describe their immunohistochemical markers, switch to filipin histochemistry and then go back to immunohistochemical and immunoblots. This should be modified.

We restructured this section in lines 222-244 to begin with a description of the cerebellar pathology, then discuss lipid accumulation, and then back to immunoblots for clearer flow. Each paragraph in this section now describes the marker and why it is being evaluated.

- Lines 209-246, Figs. 3A, 4A: The histochemical staining with filipin is an important element, but without a quantitative assessment of fluorescence intensities across different animals, the robustness and reproducibility of the data remains unclear.

Quantification of filipin is problematic in tissue due to rapid photobleaching and variability inherent to different sections and runs. Therefore, we rely on a qualitative assessment and shore up the observed minor differences in lipid accumulation with the mass spectrometry imaging shown in Fig S5 and the results section for sphingolipid accumulation (lines 256-272).

- Lines 211,212 & 233,234: The authors should consider to describe the marker CD68 in a more concise manner. Their statements that CD68 labels microglia and myeloid cells together with the sentence "..., many of which are macrophages like Kupffer cells..." can confuse the non-expert reader.

CD68 now refers to reactive microglia in the brain (lines 243-244, 330-331, 394-395) or macrophages in the liver (lines 247-248, 347-348, 411).

- Lines 214-216: Despite repeated statements in the literature, GFAP cannot be considered a marker of neuroinflammation (see Escartin et al., 2021 Nature Neurosci). An increase of the protein suggests the presence of reactive astrocytes. Notably, in the cerebellum, most of the GFAP-positive cells are probably reactive Bergmann glia as indicated by their location and shape (Fig. 3, higher mag images). Corresponding statements should be corrected.

Reactive astrocytosis is now used in lines 229-230 and corresponding statements have been corrected.

- Lines 226-231 and Fig. 3F, G: These results do not seem to fit in here. They should be described in the paragraph lines 177-203 and figure 2, where similar results are presented for cerebrum and liver.

Parallel DNA and protein measurements for cerebrum, cerebellum, and liver are now included together in Figure 2.

- Lines 243-247: The meaning of this paragraph is unclear. Up to this point, only dose dependence was described, but no data on age or the hypomorphic model.

These lines have been removed.

- Line 249-253: The title and introductory statements should mention right away in which organ or tissue the lipid distribution was analysed.

Title and introductory statement now refer to lipid distribution analysis in the brain in lines 254-258.

- Lines 249-265: Without proper quantification, the relevance of the data is unclear. As is, the dataset appears as dispensable. How many animals and sections per animal were analysed? The figure number S8 is not in line with the figures mentioned in the previous chapter.

Figure numbers were corrected so that this is now Figure S5. Number of sections and animals sample size are included in the figure legend. We acknowledge the lack of quantitative data with respect to storage of unesterified cholesterol. To orthogonally confirm the observed qualitative differences in filipin labeling, we opted to use mass spectrometry imaging of different lipid species in the brain. While there are trends within the lipid species such that increasing gene therapy doses result in lipid levels closer to those in *Npc1^{+/+}* mice, the results are not significant. Therefore, we opted to keep the mass spectrometry data as a supplemental figure. Also, people with limited familiarity of NPC disease will more easily identify pathology associated with unesterified cholesterol than changes in less well-known lipid species such as hexosyl- or dihydro-ceramides.

We have added text related to this limitation and options for more in depth analysis of lipid abnormalities in the discussion section (lines 496-503).

- Lines 260-264: The meaning of these statements is unclear. They should be rewritten. This section has been edited for clarity, lines 265-272.

- Line 268: The title appears long and verbose. The authors should consider to modify. The term describing the viral construct is used in an inconsistent manner in the different chapter titles. Viral construct naming is now consistent and the section title has been edited for clarity (lines 275-276).

- Line 319: Is there a specific reason why the markers used to detect dose effects in the cerebellum were not used to reveal age-dependent differences in this brain area? This data has been added in Figure 6 and lines 324-332.

- Line 322: Replace "In the liver, higher copy numbers predicted" by "Higher copy numbers in liver cells predicted". Edited text as suggested (lines 345-346).

- Lines 303-327: The description of the results is not well structured. As outlined for other paragraphs, the authors should tidy the text. Text has been edited to increase clarity and flow (lines 337-349).

- Lines 310-312 & 323-325, Fig. 6, S3A: The copy numbers seem to be higher the shorter the interval between vector administration and measurement. Does this mean that copy numbers decrease with time following injection?

This is an interesting question that we did not specifically evaluate. It could mean that copy numbers decrease with time following injection as the reviewer posits. It may also indicate that as neurodegeneration progresses, fewer cells are available for transduction or that cells already committed to a cell death pathway at time of transduction are lost over this timeframe, thus leading to an apparent inverse correlation between time since injection and copy number. To answer this question, a time duration post-injection matched analysis would be insightful (i.e. copy number analysis in mice treated at 4, 6, or 8 weeks and sacrificed 3 weeks post injection) as suggested by Reviewer 1. These points are included as future directions in the discussion (lines 482-489).

- Line 349: The author mention "untreated mice". Do their cohorts include any untreated = non-injected animals? This should be clarified.

Corrected throughout manuscript to clearly indicate saline-injected mutant mice and untreated wildtype mice.

- Line 353: "though were still": a subject is missing.

This has been removed.

- Lines 363-368: The staining of myeloid cells cannot indicate "liver lipid storage burden" or cholesterol accumulation. This should be corrected. The statements obviously raise the question whether the authors performed filipin staining in these animals.

This statement has been corrected to just describe macrophage labeling in 410-412.

- Line 393: Define "RUSP"!

RUSP is no longer referred to in the introduction and discussion.

- Line 371-422: Similar as for the Results, the authors should consider to revise the Discussion. As is, it appears as a bit patchy with unnecessary repetition. Moreover, some topics are not really discussed in the light of previous studies. Just as examples: lines 390-391 stand isolated, repeating what has been written in lines 380-381 and lines 394-395 mention the goal that has already been presented before. The discussion of gene copy number and outcome is superficial. Can this be compared to previous preclinical studies on NPC or other diseases? What do the present results mean for preclinical R&D in other diseases with defects in lysosomal membrane-spanning proteins?

We thank you for the helpful suggestion. Repetition has been removed and comparisons are made to clinical studies and previous NPC1 gene therapy work in the discussion (lines 425-426, 436-438, and 449-457. Broader implications regarding dose and age of treatment are discussed for other diseases with lysosomal membrane-spanning protein deficiencies in lines 515-519.

- Lines 488, 493 & 552: The authors mention "spleen, kidney, lung, and leg muscle" as well as "brainstem", but no data are shown for these organs and brain region.

These tissues are included in Supplemental Figure 2.

- Lines 607-685: The authors should revise the figure legends. A few examples:

- - Line 634: The title appears as imprecise not referring to the actually organ or tissue.
- - Line 636: Change to "astrocytes and Bergmann glia".
- - Lines 638: The IBA1 staining is mentioned in the legend, but not in the corresponding chapter of the Results. Can IBA1 really be considered a surrogate marker for neuroinflammation? Again, the authors should introduce the marker properly.
This has been edited to refer to correct tissue, reactive astrocytes and Bergmann glia (line 739).
The figure includes the compound or antibody used to visualize each lipid or protein (filipin, GFAP/Calbindin D/IBA1, respectively).

- Figures: The authors should homogenize the design and size of graphs and to arrange panels reducing white space.
Figures are more consistent and have been redesigned.

- Fig. 1: The authors should consider to remove the table from the figure. The statistics can easily be integrated in panel A by asterisks. The number of animals is already evident from panel A.
Asterisks were difficult to incorporate as was the number of animals in panel A, but the table has been moved to a supplemental table (Table S2A).

- Fig. 2: The rectangles surrounding panels should be removed. Moreover, the y axis scaling in panel Aiii should be modified (max at 0.1) to show the differences between treated groups. In addition, the authors should consider to move the representative Western blots from Fig. S5 to this main figure.
Figure has been restructured and the y-axis has been changed. Due to volume of data are contained in the main figures, the representative western blots remain as supplemental figures.

- Fig. 2C: It is unclear what the lines represent? Linear regression with 2 data points seems to make little sense. Where are green data points representing high dose? What is the difference to the graph shown in panel 2Aii?
Figure 2 has been modified to facilitate ease of comparison and interpretation for the reader. The lines represent the linear regression with all data points for a given treatment group. We have also ensured that all graphs have the same format of data represented for each tissue (A, cerebellum; B, cerebrum; C, liver).

- Fig. S2: The figure is not a figure, it's a table.
Thank you for pointing this out. The Table is now Table S2.

- Fig. S3. The data seem important, therefore they should be integrated in the main figure.
The supplemental liver data for age at treatment has been moved to Figure 7.

- Fig. S5: The figure contains legend text that should be removed.
Figure legend text has been removed.

February 26, 2026

RE: Life Science Alliance Manuscript #LSA-2024-02874R-A

Dr. Cristin D Davidson
Eunice Kennedy Shriver National Institute of Child Health and Human Development
National Human Genome Research Institute
9000 Rockville Pike
CRC Bldg 10, Rm E1-3288
Bethesda, MD 20892

Dear Dr. Davidson,

Thank you for submitting your revised manuscript entitled "Optimization of systemic AAV9 gene therapy in Niemann-Pick disease, type C1 mice". We returned this work to Reviewers 1 and 3 who are now satisfied with no further requests. We would be happy to publish your paper in Life Science Alliance pending final revisions necessary to meet our formatting guidelines.

MANUSCRIPT ORGANIZATION AND FORMATTING:

To avoid unnecessary delays in the acceptance and publication of your paper, please read the following information carefully. Full guidelines are available on our Instructions for Authors page, <https://www.life-science-alliance.org/authors>

- Please upload your Tables in editable .doc or Excel format.
- Please upload your main manuscript text as an editable doc file.
- Please add the X and Bluesky handles of your host institute/organization, as well as your own, and/or one of the authors, in our system.
- Mark the 2ndary Corresponding Author on the manuscript file as well.
- Please add your main, supplementary figure, and table legends to the main manuscript text after the references section.
- Please mark panel D in Figure 1.
- There are call-outs for figure S2C and D, and this figure doesn't have panels at all. Please correct. Same for figure S7. IT doesn't have panels F, G, and H, but there are call-outs in the manuscript text.
- Please add callouts for Figures S1C and S3A-F to your main manuscript text.
- Please include a "Data Availability" section after the Materials & Methods section. This section must indicate the availability of data from mass spectrometry imaging. Please consult our guidelines at <https://www.life-science-alliance.org/manuscript-prep#format>
- Please add an Author Contributions section to your main manuscript text.
- Certain experimental methods lack important details and instead refer readers to previously published work. Please provide full details for the following in the Methods section: Phenotypic Assessment (describe how each of the 5 behaviors was scored), Image capture and analysis (include details on the microscope and objectives), Quantification of CD68 area, Copy number analysis by ddPCR (describe this method and include primer sequences).

We welcome submissions of potential cover images for the issue of LSA in which your work would appear. If you have high quality images associated with this work, please feel free to email these, with a caption, to the journal office.

LSA encourages authors to provide a 30-60 second video where the study is briefly explained. We will use these videos on social media to promote the published paper and the presenting author (for examples, see <https://docs.google.com/document/d/1-UWCfbE4pGcDdcgzcmiuJl2XMBJnxKYeqRvLLrLSo8s/edit?usp=sharing>). Corresponding

or first-authors are welcome to submit the video. Please submit only one video per manuscript. The video can be emailed to contact@life-science-alliance.org

FINAL FILES:

The following items are required for acceptance.

The license to publish form must be signed before your manuscript can be sent to production. A link to the license to publish form will be available to the corresponding author only. Please take a moment to check your funder requirements.

Thank you for your attention to these final processing requirements. Please revise and format the manuscript and upload materials as soon as you are able.

Thank you for this interesting contribution to the literature. We look forward to publishing your paper in Life Science Alliance.

Sincerely,

Reviewer #1 (Comments to the Authors (Required)):

The authors have addressed reviewer comments and the manuscript is improved in clarity and accuracy.

Reviewer #3 (Comments to the Authors (Required)):

The authors have addressed the numerous comments/suggestions in a highly satisfactory manner. Their work is very important work for the field.

March 17, 2026

RE: Life Science Alliance Manuscript #LSA-2024-02874RR

Dr. Cristin D Davidson
Eunice Kennedy Shriver National Institute of Child Health and Human Development
National Human Genome Research Institute
9000 Rockville Pike
CRC Bldg 10, Rm E1-3288
Bethesda, MD 20892

Dear Dr. Davidson,

Thank you for submitting your Follow Up entitled "Optimization of systemic AAV9 gene therapy in Niemann-Pick disease, type C1 mice". It is a pleasure to let you know that your manuscript is now accepted for publication in Life Science Alliance. Congratulations on this interesting work.

Your article will publish open access upon publication under a CC-BY license.

DISTRIBUTION OF MATERIALS:

Again, congratulations on a very nice paper. I hope you found the review process to be constructive and are pleased with how the manuscript was handled editorially. We look forward to future exciting submissions from your lab.

Sincerely,
